# Effects of myosin variants on interacting-heads motif explain distinct hypertrophic and dilated cardiomyopathy phenotypes

Lorenzo Alamo[1†], James S Ware[2,3,4†], Antonio Pinto[1], Richard E Gillilan[5], Jonathan G Seidman[4], Christine E Seidman[4,6*†], Raúl Padrón[1*†]

[1]Centro de Biología Estructural, Instituto Venezolano de Investigaciones Científicas, Caracas, Venezuela; [2]National Heart and Lung Institute and MRC London Institute for Medical Sciences, Imperial College London, London, United Kingdom; [3]NIHR Cardiovascular Biomedical Research Unit, Royal Brompton and Harefield NHS Foundation Trust and Imperial College London, London, United Kingdom; [4]Department of Genetics, Harvard Medical School, Boston, United States; [5]Macromolecular Diffraction Facility, Cornell High Energy Synchrotron Source, Ithaca, United States; [6]Cardiovascular Division, Brigham and Women's Hospital and Howard Hughes Medical Institute, Boston, United States

*For correspondence: cseidman@genetics.med.harvard.edu (CES); raul.padron@gmail.com (RP)

†These authors contributed equally to this work

**Abstract** Cardiac $\beta$-myosin variants cause hypertrophic (HCM) or dilated (DCM) cardiomyopathy by disrupting sarcomere contraction and relaxation. The locations of variants on isolated myosin head structures predict contractility effects but not the prominent relaxation and energetic deficits that characterize HCM. During relaxation, pairs of myosins form interacting-heads motif (IHM) structures that with other sarcomere proteins establish an energy-saving, super-relaxed (SRX) state. Using a human $\beta$-cardiac myosin IHM quasi-atomic model, we defined interactions sites between adjacent myosin heads and associated protein partners, and then analyzed rare variants from 6112 HCM and 1315 DCM patients and 33,370 ExAC controls. HCM variants, 72% that changed electrostatic charges, disproportionately altered IHM interaction residues (expected 23%; HCM 54%, p=$2.6\times10^{-19}$; DCM 26%, p=0.66; controls 20%, p=0.23). HCM variant locations predict impaired IHM formation and stability, and attenuation of the SRX state - accounting for altered contractility, reduced diastolic relaxation, and increased energy consumption, that fully characterizes HCM pathogenesis.

## Introduction

Vertebrate skeletal and cardiac muscles produce force through a chemo-mechanical cycle that contracts and relaxes the sarcomere, the contractile unit in all muscle cells. Sarcomeres are composed of inter-digitating thick filaments that contain myosin, comprising a globular head (subfragment 1; S1) and α-helical tail, and thin filaments that contain actin. Interactions between myosin and actin generate the contractile force underpinning sarcomere contraction.

Myosin heads comprise a motor domain (MD) that hydrolyses ATP, a regulatory domain (RD) that includes the essential (ELC) and regulatory (RLC) light chains, and a proximal S2 subfragment. Myosin tails are packed in antiparallel formation and shape the thick filament's backbone, from which myosin heads protrudes every 14.3 nm as crowns formed by three pairs of heads. In some crowns (*Zoghbi et al., 2008*) the S2 fragment interacts with myosin binding protein-C (MyBP-C), another thick filament protein that modulates sarcomere activity. Contraction occurs through conformational

changes in the myosin MD that accompany ATP hydrolysis, increasing affinity for actin in the thin filament, and resulting in a powerstroke that transfers force by rotation of the myosin lever domain around the converter domain.

After contraction, myosin heads rebind ATP, detach from actin, and assume a relaxed pre-powerstroke state (*Llinas et al., 2015*). During relaxation, two myosin heads pack together to form an interacting-heads motif (IHM; [*Woodhead et al., 2005*]). Consecutive IHMs interact along helices that protrude from the thick filament backbone and limit potential interactions between myosin heads and actin (*Alamo et al., 2008*; *Jung et al., 2008*). The IHM forms only in relaxed muscle and is a highly conserved structure throughout the evolution of myosins among different species, in different muscle types (smooth, skeletal, and cardiac), and non-muscle myosins (*Alamo et al., 2016*).

The IHM appears to form through key interactions (*Figure 1A*) involving the actin-binding region of one myosin head (denoted as 'blocked') and the converter region of the partner 'free' head (*Woodhead et al., 2005*; *Alamo et al., 2008*; *Jung et al., 2008*). IHM priming occurs when the blocked head assumes a pre-powerstroke state (*Llinas et al., 2015*) and docks onto its own S2 by

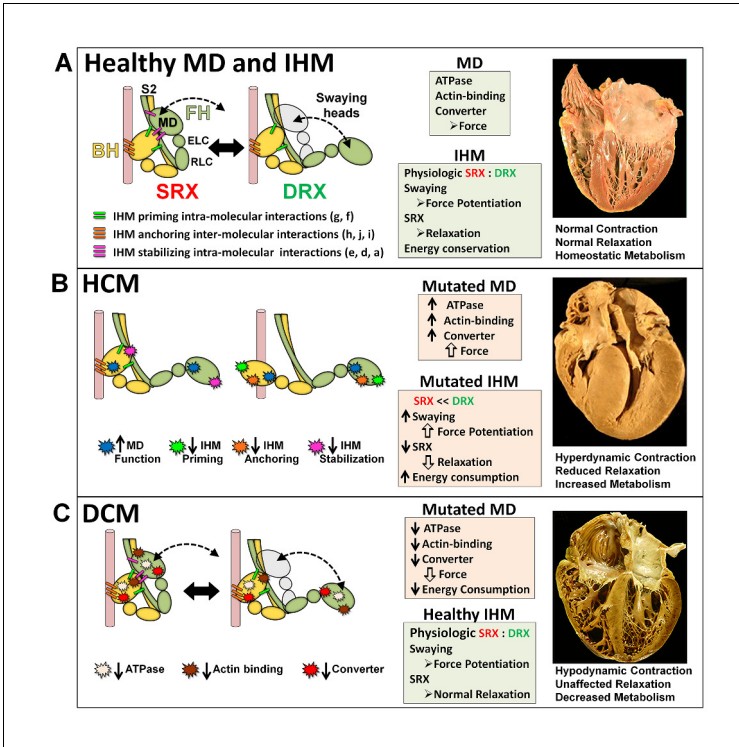

**Figure 1.** The molecular pathogenesis of hypertrophic (HCM) and dilated (DCM) cardiomyopathy assessed in the context of the myosin interacting-heads motif (IHM) paradigm. Myosin interactions involved in IHM assembly and myosin motor domain (MD) functions that are altered by PVs and LPVs are depicted. (**A**) Relaxed healthy cardiac muscle contains myosin heads populations in the super-relaxed (SRX) state (left) with lowest ATP consumption and a disordered relaxed (DRX) state (right) with swaying free heads that generate force with higher ATP consumption. The population of cardiac myosins in SRX is more stable than in skeletal muscle (*Hooijman et al., 2011* and see Material and methods) which supports physiologic contraction and relaxation, energy conservation, and normal cardiac morphology. (**B**) HCM myosin variants both alter residues involved in MD functions (causing increased biophysical power [*Tyska et al., 2000*]), and destabilize IHM interactions (particularly those with altered electrostatic charge). Reduced populations of myosins in the SRX state and increased populations of myosins in DRX as well as enhanced MD properties will result in increased contractility, decreased relaxation, and increased ATP consumption, the three major phenotypes observed in HCM hearts. Compensatory signals may promote ventricular hypertrophy. (**C**) *MYH7* DCM variants have modest effects on IHM interactions but substantially reduce MD functions, particularly nucleotide binding, resulting in reduced ATP consumption and sarcomere power (*Schmitt et al., 2006*), with minimal impact on relaxation and overall diminished contractility. Compensatory signals result in ventricular dilatation to maintain circulatory demands in DCM hearts.

intra-molecular interactions. IHM anchoring occurs by inter-molecular interactions between the docked, blocked head and the neighboring myosin's tail, and intra-molecular interactions with its ELC. IHM stabilization occurs by intra-molecular interactions between the free myosin head and the anchored blocked head (*Alamo et al., 2008*; *Brito et al., 2011*; *Alamo et al., 2016*). IHM scaffolding interactions between myosin heads and both ELCs and RLCs further support its three-dimensional shape.

The dynamic structure of the IHM (*Brito et al., 2011*) enables properties that are not observed in isolated S1 myosin fragments (*Figure 1A*): (i) reduced ATPase activity in blocked heads and intermittently in the free head; (ii) mobility of the free head, that sways forth and back by breaking and restoring stabilizing IHM interactions (*Alamo et al., 2008*; *Brito et al., 2011*; *Sulbarán et al., 2013*; *Espinoza-Fonseca et al., 2015*; *Alamo et al., 2015*; *Yamaguchi et al., 2016*; *Alamo et al., 2016*); and (iii) release of extra free heads upon activation and blocked heads for force potentiation (*Brito et al., 2011*). Consistent with these properties, negatively stained smooth muscle myosins show that free heads are flexible, detaching from their partner blocked heads and adopting different motor domain orientations (*Burgess et al., 2007*); that swaying free heads support the sliding of F-actin filaments along relaxed thick filaments (*Brito et al., 2011*); and that thick filament respond to calcium activation producing force immediately (*Linari et al., 2015*).

The asymmetric IHM structure imposes functional consequences on the sarcomere. The blocked myosin head has a pre-powerstroke state (*Xu et al., 2003*; *Zoghbi et al., 2004*; *Llinas et al., 2015*) while docked onto its S2, so ATPase activity is sterically inhibited, and the actin-binding interface is positioned on the converter domain of the free head, so that it cannot bind actin. By contrast, the partnered free myosin head can sway, hydrolyze ATP, and bind actin, producing disordered relaxation (DRX; [*McNamara et al., 2016*]). In addition, the asymmetric conformations of myosins could also impact RLC phosphorylation, which modulates the rate and extent of force development in cardiac muscle (*Toepfer et al., 2016*). The RLC associated with the free head is more accessible to protein kinase C phosphorylation (*Alamo et al., 2008*), which promotes swaying (*Brito et al., 2011*) and potential actin interactions.

The IHM structure could account for super-relaxation (SRX; [*Stewart et al., 2010*]). When the free myosin head docks onto the blocked head, its ATPase is also inhibited through stabilization of converter domain movements that are needed to achieve strong actin binding (*Wendt et al., 2001*). The dual enzymatic inhibition of SRX is reflected by a low ATP turnover rate, detected by quantitative epi-fluorescence in skeletal (*Stewart et al., 2010*; *Naber et al., 2011*; *McNamara et al., 2016*) and cardiac (*Hooijman et al., 2011*) relaxed muscle fibers, and is proposed as an energy saving mechanism (*Cooke, 2011*). Muscle relaxation can therefore be characterized by two myosin head IHM conformations and ATPase activities (*Figure 1A*): the DRX state with swaying free myosin heads and ATPase rate comparable to that observed in single myosin molecules, and the SRX state with a highly inhibited (five-fold lower) ATPase rate (*Stewart et al., 2010*; *Naber et al., 2011*; *McNamara et al., 2016*; *Hooijman et al., 2011*).

Hypertrophic (HCM) and dilated (DCM) cardiomyopathies are heart muscle diseases caused by variants in genes encoding sarcomere proteins (*Geisterfer-Lowrance et al., 1990*; *Seidman and Seidman, 1991*; *Konno et al., 2010*), most commonly *MYH7* (encoding cardiac $\beta$-myosin heavy chain; MHC) and *MYPBC3* (encoding MyBP-C) and rarely in *MYL2* and *MYL3* that encode the ELC and RLC, respectively. Distinct variants in sarcomere genes also cause DCM, albeit less frequently. Variant location is not highly predictive of whether it will trigger HCM or DCM (*Colegrave and Peckham, 2014*).

HCM hearts exhibit hyperdynamic contraction, impaired relaxation, and increased energy demands (*Ashrafian et al., 2011*). Impaired relaxation (clinically denoted as diastolic dysfunction) is a major cause of HCM symptoms and arrhythmias. Increased energy demands in HCM hearts are thought to contribute to cardiomyocyte death and progression to heart failure. Previous studies demonstrate that HCM *MYH7* variants increase myofibrillar $Ca^{2+}$-sensitivity and sarcomere power, therein explaining hyperdynamic contraction (*Seidman and Seidman, 2001*; *Tyska et al., 2000*; *Moore et al., 2012*; *Spudich, 2014*; *Marston, 2011*; *Harris et al., 2011*; *Ashrafian et al., 2011*; *Spudich et al., 2016*), but the mechanisms accounting for diminished relaxation and excessive energy requirements remain poorly understood. By contrast, DCM *MYH7* variants decrease $Ca^{2+}$-sensitivity and reduce motor domain functions (*Debold et al., 2007*; *Moore et al., 2012*; *Spudich, 2014*; *Marston, 2011*; *Spudich et al., 2016*), therein accounting for diminished contractile

performance, the prototypic feature of DCM. Unlike HCM, diastolic dysfunction is uncommon early in the clinical course of DCM (*Lakdawala et al., 2012*) but emerges with myocardial remodeling.

Disease-associated *MYH7* variants cause single amino acid (aa) substitutions, and as such the locations of these substitutions inform physiologically important domains in myosin. Because myosin sequences are highly conserved across species, the impact of *MYH7* variants on MD properties have been studied in the context of a variety of vertebrate molecular myosin structures. Early analyses, employing the chicken skeletal S1 myosin (*Rayment et al., 1995*, *1993*), demonstrated that HCM variants perturbed MD residues involved in nucleotide-binding, actin-binding, or the relay, converter, and lever. More recently additional HCM variants were located onto the myosin mesa (*Homburger et al., 2016*), a newly characterized flat broad (>20 nm$^2$) structural feature that is proposed to bind MyBP-C (*Spudich, 2015*; *Spudich et al., 2016*). The mesa is prominent in the pre-stroke S1 contractile state, but substantially reduced following the powerstroke (*Homburger et al., 2016*).

The tarantula striated IHM quasi-atomic model PDB 3DTP (*Alamo et al., 2008*) complements these structures by defining interactions between adjacent myosin heads and possible interactions with IHM-associated protein partners and the backbone. Harnessing this model with other structures (human S2 crystal PDB 2FXM model [*Blankenfeldt et al., 2006*] and chicken smooth muscle myosin PDB 1I84 model [*Liu et al., 2003*]), the analyses of a few selected HCM variants (*Blankenfeldt et al., 2006*; *Alamo et al., 2008*; *Moore et al., 2012*) hint that these may impact residues involved in IHM intra- and intermolecular interactions.

Here we assessed the structural pathogenesis of HCM and DCM by comprehensive analyses of the largest series of genotyped cardiomyopathy cases, comprising 7427 individuals with clinical-grade sequencing (*Walsh et al., 2017*). Modeling a human homologous *β*-cardiac myosin IHM model (PDB 5TBY, newly reported here) to the newest tarantula IHM quasi-atomic structure PDB 3JBH (*Alamo et al., 2016*), we first identified putative interaction sites, then examined the locations of all variants categorized as pathogenic (PVs) and likely pathogenic (LPVs) in *MYH7*, *MYL2*, and *MYL3*. We demonstrate that HCM and DCM variants differentially impact myosin MD functions and specific IHM interactions, data that informs the mechanisms for failed contractile power in DCM and the markedly compromised relaxation and energetics in HCM (*Ashrafian et al., 2003*, *2011*).

## Results

### Human β-cardiac myosin IHM quasi-atomic PDB 5TBY model to identify interaction sites

We modeled a homologous human *β*-cardiac myosin IHM structure (*Figure 2A* and *Video 1*; deposited as entry PDB 5TBY) based on the tarantula striated muscle IHM quasi-atomic cryo-EM structure (PDB 3JBH, *Alamo et al., 2016*; Materials and methods). Although resolution of the IHM structure as derived by cryo-EM does not define specific atomic contacts or individual side chain densities, this model nonetheless identifies molecular interactions.

### Comparisons of the human *β*-cardiac myosin IHM quasi-atomic PDB 5TBY with existing models

The new model fitted well to the 28 Å resolution 3D-reconstruction for the negatively stained human cardiac thick filament (EMD-2240)(*Al-Khayat et al., 2013*) (*Figure 2A*). In addition, the predicted SAXS profile of PDB 5TBY is consistent with X-ray solution scattering of squid HMM in Ca$^{2+}$-free (EGTA) conditions (*Figure 2B* and *Figure 2—figure supplement 1*) and the computed scattering profile for the model closely agrees with the measured scattering profile of squid heavy meromyosin (HMM) (*Gillilan et al., 2013*). *Figure 2C* shows the residuals (relative deviation) between the squid HMM data and the computed 5TBY scattering profile over the range of scattering angles (red line). The χ goodness of fit parameters reported by the FoXS program are 1.5 for 3JBH and 1.85 for 5TBY respectively (*Schneidman-Duhovny et al., 2013*). Relative deviation between the 5TBY and 3JBH calculated scattering profiles (dashed line) falls below the noise level of the experimental data (red line) consequently the two models cannot be distinguished. Extending the computations to wider angles (q = 1.0 Å$^{-1}$; see *Figure 2B* and *Figure 2—figure supplement 1*), revealed that the tarantula PDB 3JBH and the human models are essentially identical above q = 0.4 Å$^{-1}$ and unlikely to be

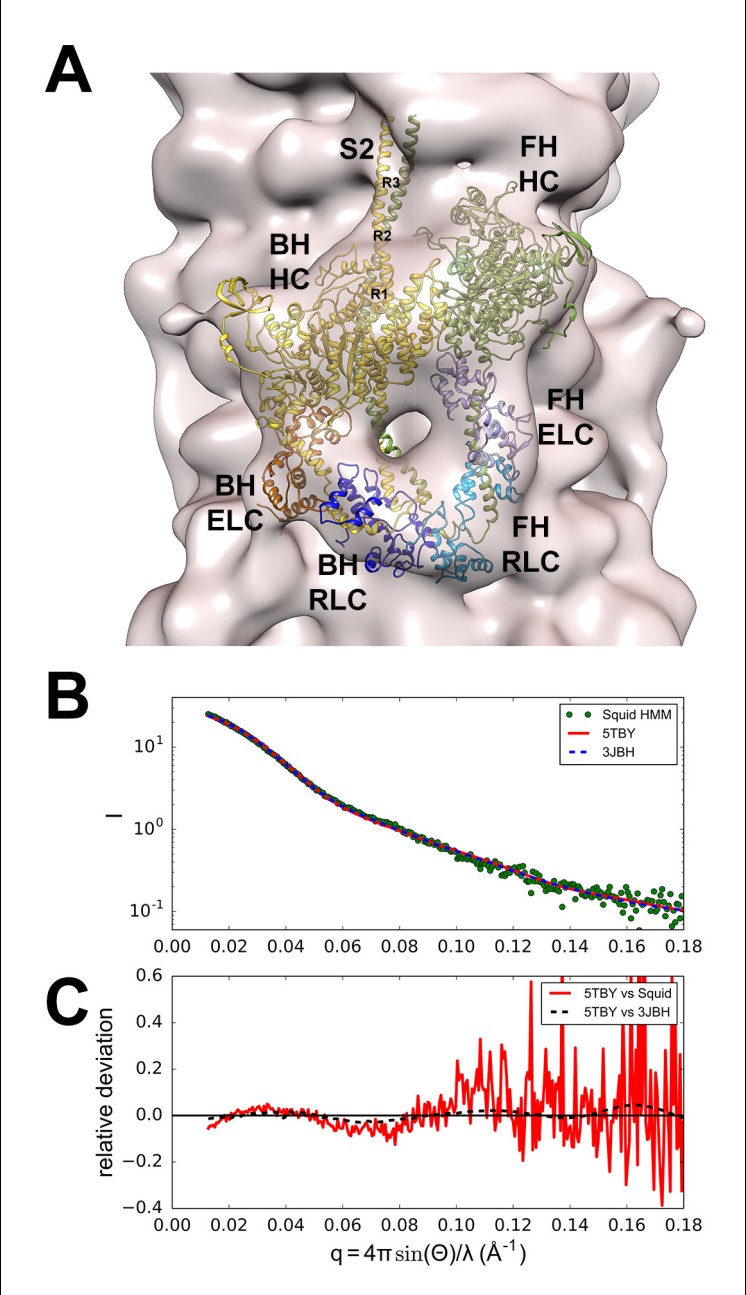

**Figure 2.** Structure of the human β-cardiac myosin interacting-heads motif. (**A**) Quasi-atomic homologous model of human β-cardiac myosin interacting-heads motif (IHM) PDB 5TBY composed of blocked (BH) and free (FH) heads, fitted to the human cardiac thick filament 3D-map EMD-2240 (*Al-Khayat et al., 2013*). (Also see *Video 1*.) Domains and residue equivalence are provided in *Supplementary file 1*. Sarcomere proteins depicted are MHC (BH: gold, FH: olive), essential light chain (ELC associated with BH: brown, FH: purple), and regulatory light chain (RLC associated with BH: dark blue, FH: blue). The three negatively-charged rings in the S2 are labeled R1, R2 and R3. (**B**) Calculated small angle X-ray solution scattering (SAXS) profile of PDB 5TBY (red line) matches the experimental squid heavy meromyosin SAXS profile (green dots) (*Gillilan et al., 2013*). Integrated scattering intensity (I in arbitrary units) is given as a function of momentum transfer, $q = 4\pi \sin(\theta)/\lambda$, with a scattering angle of $2\theta$ and a wavelength of $\lambda$. (**C**) Relative deviation between PDB 5TBY scattering and squid HMM (red line) is calculated as $(I_{model}-I_{exp})/I_{exp}$. The corresponding difference in scattering between PDB 5TBY and PDB 3JBH models is also shown on the same scale (black dashed line). Comparison shows that the models cannot be distinguished based on currently available scattering data (See Supplementary Figures).

*Figure 2 continued on next page*

*Figure 2 continued*

The following figure supplements are available for figure 2:

**Figure supplement 1.** Calculated small angle X-ray solution scattering (SAXS) profile of PDB 5TBY (red line) matches experimental squid heavy meromyosin (HMM) SAXS profile (green dots) (*Gillilan et al., 2013*).

**Figure supplement 2.** Wide-eye stereo pairs that compare the PDB 5TBY model with crystal structures of fragments for human *β*-cardiac S1 MD fragment (PDB 4DB1) and S2.

distinguished by future SAXS and wide angle X-ray solution scattering experiments in that regime. The SAXS data in this angle range provide support for the overall shape, placement, and orientation of IHM domains.

The PDB 5TBY model also closely matches crystal structures of fragments for human *β*-cardiac S1 MD fragment (PDB 4DB1) (*Figure 2—figure supplement 2A* and *Video 2*) and S2 (PDB 2FXM) (*Blankenfeldt et al., 2006*) (*Figure 2—figure supplement 2B* and *Video 3*). Moreover, this model includes important myosin loops that improve the analyses of IHM interactions, and that are absent from the only other IHM model used for cardiac muscle (PDB 3DTP; *Alamo et al., 2008*). In the PDB 5TBY model, the blocked and free myosin heads are in the pre-powerstroke state (*Xu et al., 2003*; *Zoghbi et al., 2004*; *Llinas et al., 2015*) and bent at the 'pliant region' (*Houdusse et al., 2000*) (see *Figure 2—figure supplement 2C* and *Video 4*). The blocked head is in a pre-powerstroke state (*Llinas et al., 2015*) (similar to the Mg·ADP-AlF$_4$, PDB 1BR1; [*Liu et al., 2003*; *Alamo et al., 2008*]) while the free head, which is also in a pre-powerstroke state, shows a less angled lever arm than the blocked head (see *Figure 2—figure supplement 2C*). In the absence of a human *β*-cardiac IHM crystal atomic structure, the homologous human *β*-cardiac myosin IHM model PDB 5TBY (*Figure 2A* and *Video 1*) provides a robust option for analyses of cardiomyopathy variants on IHM interactions.

## Analyses of HCM and DCM variants

All sarcomere PVs and LPVs previously identified in 6112 HCM and 1315 DCM cases (*Walsh et al., 2017*) were analyzed. Among HCM cases, 40 *MYH7* PVs substituted 39 distinct aa (two distinct variants encoded an identical substitution) at 33 positions and 95 LPVs led to 95 substitutions at 87 sites (*Tables 1* and *2*). Thirty-eight PVs altered S1 and S2 residues, and two altered LMM residues. Four further HCM PVs occurred in ELC (n = 2) or RLC (n = 2). Among DCM cases, one *MYH7* PV and 26 LPVs altered 26 distinct aa residues (*Table 3*). We considered how these variants impacted residues involved in motor domain functions (*Supplementary file 1*) and in pre-specified IHM interactions (*Supplementary file 2*).

## Variants altering functional Motor Domain residues

We compared the distributions of *MYH7* HCM and DCM variants at functional MD residues involved in the nucleotide-binding pocket (39 aa), actin binding (60 aa), converter (68 aa), or

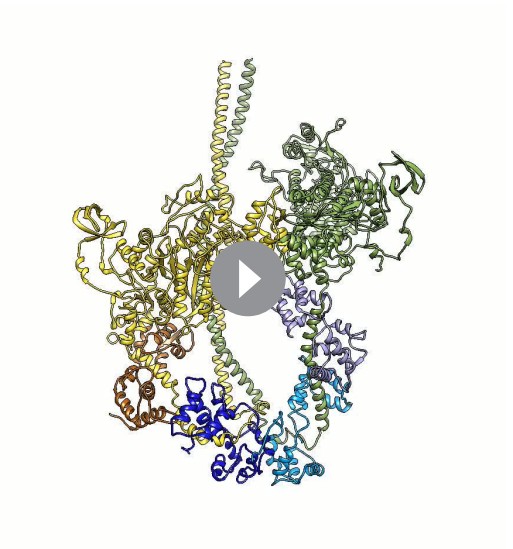

**Video 1.** A homologous human *β*-cardiac myosin IHM structure. The PDB 5TBY IHM model is initially depicted as a ribbon structure that includes paired myosin heads (blocked head (BH), gold, free head (FH), olive), essential light chain (ELC associated with BH: brown, FH: purple), and regulatory light chain (RLC associated with BH: dark blue, FH: blue). The PDB 5TBY IHM model is then fitted into the human cardiac thick filament (3D-map EMD-2240; A(*Al-Khayat et al., 2013*) as detailed in Materials and methods. See legend of *Figure 2A*.

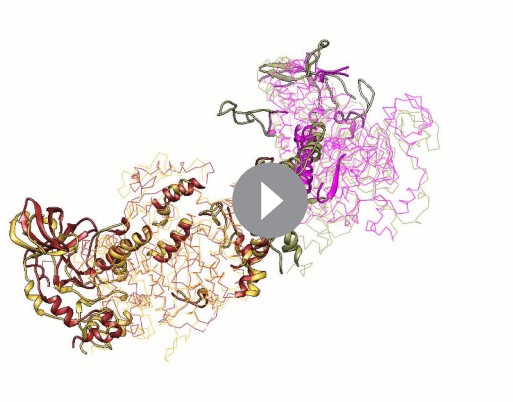

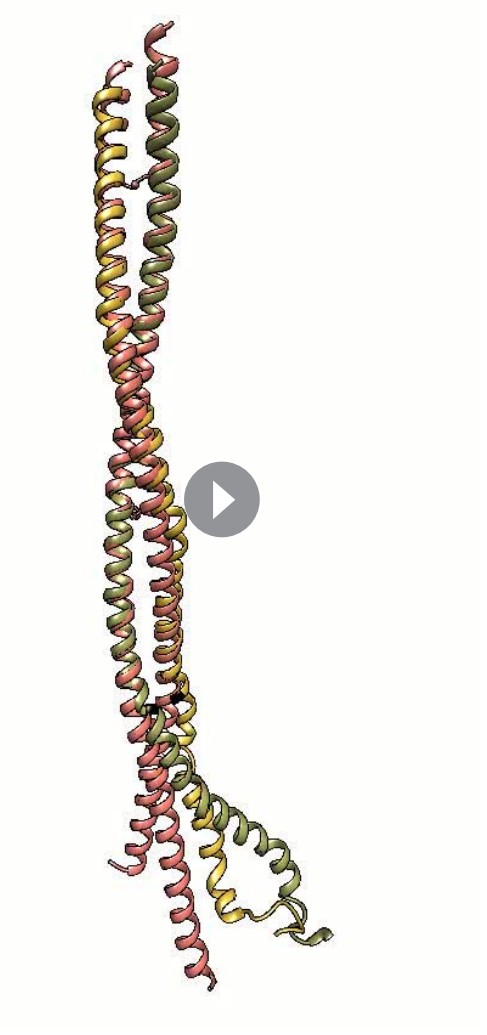

**Video 2.** The motor domain (MD) of the PDB 5TBY model closely matches the crystal structure of a fragment for human *β*-cardiac S1 MD (PDB 4DB1). The movie shows superimposed MDs of S1s from the PDB 5TBY model and a S1 fragment of PDB 4DB1 (S1 bound to adenylylimidodiphosphate (AMPPNP), a nonhydrolysable analogue of ATP to induce a rigor-like structure, open conformation) as ribbons. The MDs of PDB 5TBY blocked and free heads are in gold and olive respectively while the corresponding fragments from PDB 4DB1 are in brown and magenta. See legend of *Figure 2—figure supplement 2A*.

**Video 3.** The sub-fragment 2 (S2) of the PDB 5TBY IHM model closely matches the crystal structures of S2 for human *β*-cardiac (PDB 2FXM)(*Blankenfeldt et al., 2006*). The movie shows the two S2 strands as ribbons of PDB 5TBY in gold and olive (blocked and free heads, respectively) and 2FXM in pink. See legend of *Figure 2—figure supplement 2B*.

relay (27 aa), that together make up 10% of the myosin heavy chain (194 aa; *Supplementary file 1*). *MYH7* variants were not distributed uniformly, but were over-represented at functional MD residues both in HCM (16/40 PVs (40%), 4-fold enrichment above 10% expected, p=6.14e-07; 23/95 LPVs (24%), p=5.11e-05; combined 39/135 (29%), p=7.9e-9; *Tables 1*, *2* and *4*) and DCM (7/27 variants (26%), p=0.022; *Tables 3* and *5*). However these impacted distinct regions. HCM variants were enriched in residues of the converter (n = 21/135 (16%), expected 3.5%, p=1.25e-08; *Supplementary file 3*) while 5/27 DCM variants altered nucleotide-binding pocket residues (19%, expected 2.0%, p=0.00019; *Supplementary file 4*). This supports previous observations that HCM and DCM variants alter distinct functional motor domains (*Debold et al., 2007*; *Spudich et al., 2016*), leading to altered myocardial contraction (*Figure 1*).

## HCM variants altering IHM interactions

We next defined the distribution of HCM and DCM variants on 447 *MYH7* residues involved in major IHM interactions (*Supplementary file 2*; *Alamo et al., 2008*, *Alamo et al., 2016*). Priming interactions ('f' and 'g', involving 113 aa) dock the blocked head onto its own S2. Anchoring (156 aa) associates the blocked head and ELC to the neighboring myosin tail via interactions 'i', 'h', and 'j'. Stabilizing (189 aa) docks the free head onto the blocked head via interactions 'a', 'd', and 'e'. Scaffolding (120 aa) supports the IHM three-dimensional structure (RLC-MHC and ELC-MHC interactions). Additional RLC-RLC interactions regulate the IHM (*Alamo et al., 2016*). As residues can

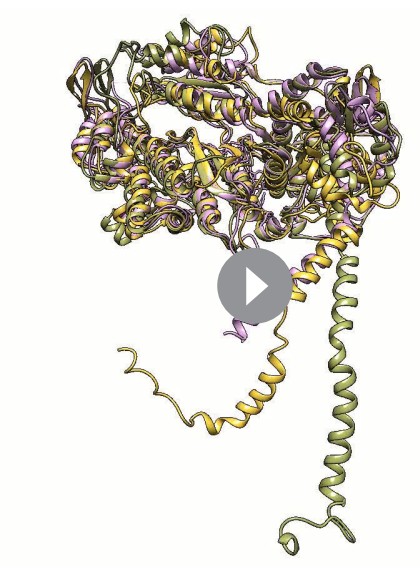

**Video 4.** The PDB 5TBY IHM model shows that the blocked and free myosin heads are in the pre-powerstroke state (*Zoghbi et al., 2004*; *Xu et al., 2003*; *Llinas et al., 2015*) and bent at the 'pliant region' as defined by *Houdusse et al. (2000)*. The movie shows the free and blocked head structures of PDB 5TBY IHM model as ribbons in gold and olive respectively, versus the pre-powerstroke state PDB 1BR1 crystal structure (magenta). The ELC and RLC were removed to highlight their lever arms, which are in the same plane but have different angles. See legend of *Figure 2—figure supplement 2C*.

participate in several interactions, HCM and DCM variants that alter multi-interactive residues can impact formation of the IHM in several ways.

In comparison to expectation, HCM PVs were enriched among the 23% of myosin residues (447 of 1935) involved in the IHM structure, with thirty-one of 40 (78%) *MYH7* PVs altering IHM residues (3.35-fold enrichment; p=5.25e-13; *Table 1 Supplementary file 3a*, *Figure 3—figure supplement 1A*). We validated this estimate by analyzing LPVs (*Table 2*): Forty-two of 95 LPVs (44%) similarly impacted IHM interactions (1.9-fold enrichment; p=7.04e-06; *Supplementary file 3b*). Based on the significant enrichment of HCM variants in residues involved in IHM interactions (*Table 4*) we infer that disruption of the IHM contributes to HCM pathophysiology (*Figure 1B*).

As cardiomyopathy variants can alter distinct interactions when located on the blocked head versus the free head, we assessed HCM PVs and LPVs exposed to these different environments (*Figure 3A* and *Video 5*, *Supplementary file 3f*). Thirty-nine of 135 PVs or LPVs alter residues involved in free head interactions (29%, expected 8.8%, p=1.93e-11), 52 altered residues involved in blocked head interactions (39%, expected 19%, p=9.32e-08), and 14 altered residues involved in tail interactions (10%, expected 3.6%, p=3.57e-04). HCM variants that disrupt priming and anchoring residues would prevent the docking of the blocked head to the S2, delaying IHM formation (*Figure 1B*, left), while variants that alter stabilizing residues will tend to release the free head from the partner blocked head

(*Figure 1B*, right). HCM PVs and LPVs (*Supplementary file 3a, b*) were particularly enriched in IHM stabilizing interactions (p=5.37e-16 vs. expectation; *Table 4*, *Figure 3—figure supplement 4*). These 48 variants (*Figure 3*, pink) include 11 variants affecting stabilizing residues when located on the blocked head (p=2.01e-2) and 31 variants affecting stabilizing residues when located on the free head (p=2.58e-12). Twenty-four HCM variants (*Figure 3A*, orange and *Figure 3—figure supplement 3*) altered blocked head residues involved in IHM anchoring interactions (p=2.1e-4). Five of eight anchoring variants (involved in 'i' interactions; *Figure 3—figure supplement 3*) affect residues that also participate in another IHM interaction (*Tables 1* and *2*), we cannot discern whether one or both interactions are more relevant to HCM pathogenesis. HCM variants were only modestly enriched in IHM priming interactions (p=2.6e-3; *Table 4*, *Figure 3A*, green and *Figure 3—figure supplement 2*), and affected nine residues on the blocked head and 14 tail residues, including three residues that also participate in stabilizing interactions (*Supplementary file 3d*). In addition, 24 *MYH7* variants (*Figure 3A*, white; *Figure 3—figure supplement 5*) altered residues that scaffold the MHC and ELC or RLC (p=3.0e-6; *Supplementary file 3d*), that were particularly enriched in MHC blocked head residues affecting ELC interactions (p=1.19e-07).

All four HCM PVs in *MYL2* (n = 2) and *MYL3* (n = 2) also affected IHM residues. The *MYL3* variants altered ELC residues involved in anchoring (orange) and scaffolding (white) the blocked head (*Table 1* and *Figure 3A*), and residues involved in stabilizing (pink) and scaffolding (white) the free head. *MYL2* variants altered residues on the RLC involved in scaffolding (*Figure 3A*, white). These variants also altered residues involved in RLC-RLC regulation (*Figure 3A*, yellow) of the blocked and free heads.

**Table 1.** HCM pathogenic variants (PVs) causing 39 *MYH7*, 2 *MYL2*, and 2 *MYL3* amino acid substitutions.

| Substitution | Location* | BH interaction* | Type* | FH interaction* | Type* | Δq† | Motor Domain Function‡ |
|---|---|---|---|---|---|---|---|
| *MYH7* | **Motor Domain** | | | | | | |
| Y115H | Near ATP Binding I | | | | | +1 | |
| E170K | Near ATP-binding II | | | d.1 | Stabilizing | +2 | |
| R249Q | Near ATP Binding III | | | | | -2 | |
| I263T | Near ATP Binding IV | | | | | 0 | |
| P307H | Near ATP Binding IV | | | | | +1 | |
| R403W/G/L/Q | CM-Loop | d.1 | Stabilizing | | | -2 | Actin interface 1 |
| R442C | Near ATP Binding V | | | d.1 | Stabilizing | -2 | |
| R453C | Near ATP Binding V | g | Priming | d.1 | Stabilizing | -2 | |
| E483K | Near Relay | j | Anchoring | d.2 | Stabilizing | +2 | |
| M515T | Relay | j | Anchoring | d.2 | Stabilizing | 0 | Relay |
| V606M | Helix-loop-helix | e | Stabilizing | | | 0 | Actin interface |
| R663H | Near Loop 2 | f.2 | Priming | | | -1 | |
| M690T | SH2 helix | | | | | 0 | |
| I702N | SH1 helix | | | | | 0 | |
| G708A | SH1 helix | | | | | 0 | |
| G716R | Converter | i | Anchoring | d.2 | Stabilizing | +2 | Converter |
| | | ELC-MHC | Scaffolding | | | | Converter |
| R719W/Q | Converter | i, | Anchoring | d.2 | Stabilizing | -2 | Converter |
| | | ELC-MHC | Scaffolding | | | | Converter |
| R723G/C | Converter | i | Anchoring | d.2 | Stabilizing | -2 | Converter |
| | | ELC-MHC | Scaffolding | | | | Converter |
| I736T | Converter | j | Anchoring | d.2 | Stabilizing | 0 | Converter |
| | | ELC-MHC | Scaffolding | | | | Converter |
| G741W / R | Converter | j | Anchoring | d.2 | Stabilizing | 0 / + 2 | Converter |
| | | ELC-MHC | Scaffolding | | | | Converter |
| K762R | Converter | j | Anchoring | d.2 | Stabilizing | +1 | Converter |
| | | ELC-MHC | Scaffolding | | | | Converter |
| K766Q | Converter | | | d.2 | Stabilizing | -1 | Converter |
| | **Regulatory Domain** | | | | | | |
| A797T | Neck-ELC interface | ELC-MHC | Scaffolding | ELC-MHC | Scaffolding | 0 | |
| F834L | Head-tail junction | RLC-MHC | Scaffolding | RLC-MHC | Scaffolding | 0 | |
| | **S2** | | | | | | |
| S842N | Head-tail junction | RLC-MHC | Scaffolding | RLC-MHC | Scaffolding | 0 | |
| R870H | Kink | g | Priming | g | Priming | -1 | |
| D906G | Ring1 | f.2 | Priming | f.2 | Priming | +1 | |
| L908V | Ring1 | f.2 | Priming | f.2 | Priming | 0 | |
| L915P | Ring2 | a | Stabilizing | a | Stabilizing | 0 | |
| E924K | Ring2 | a | Stabilizing | a | Stabilizing | +2 | |
| E930K | Ring2 | a | Stabilizing | a | Stabilizing | +2 | |
| | **Light Meromyosin** | | | | | | |
| A1379T | Near skip 2 residue | | | | | 0 | |
| R1781C | Near skip 4 residue | | | | | -2 | |
| *MYL3* | **Essential Light Chain** | | | | | | |

*Table 1 continued on next page*

*Table 1 continued*

| Substitution | Location* | BH interaction* | Type* | FH interaction* | Type* | Δq[†] | Motor Domain Function[‡] |
|---|---|---|---|---|---|---|---|
| M149V | Chain F | i | Anchoring | e | Stabilizing | 0 | |
| | | ELC-MHC | Scaffolding | | | | |
| H155D | Chain F | i | Anchoring | ELC-MHC | Scaffolding | -2 | |
| | | ELC-MHC | Scaffolding | e | Stabilizing | | |
| *MYL2* | Regulatory Light Chain | | | | | | |
| E22K | Near S15 | RLC-RLC | Regulating | RLC-RLC | Regulating | +2 | |
| | | RLC-MHC | Scaffolding | RLC-MHC | Scaffolding | | |
| R58Q | Chains B-C loop | RLC-RLC | Regulating | | | -2 | |
| | | RLC-MHC | Scaffolding | RLC-MHC | Scaffolding | | |

All PVs identified in 6112 HCM patients (**Walsh et al., 2017**) are shown. In *MYH7* there are 40 PVs, leading to 39 distinct substitutions at 33 positions while in *MYL2* and *MYL3* there are four more pathogenic variants. HCM PVs located on the mesa are bolded.

***Supplementary file 1** defines domain locations and **Supplementary file 2** defines IHM interactions, refined using the PISA analysis (**Krissinel and Henrick, 2007**) that are highlighted as in **Figure 3**, excluding the two LMM substitutions. RLC–RLC denotes interface region between regulatory light chains.

[†]Δq: variant induced electrical charge change. Δq defines whether the substituted amino acid changes the charge (bolded) or not (Δq = 0).

[‡]Motor domain functions are ascribed only to variants within involved residues. RLC–RLC denotes interface region between regulatory light chains. Variants at IHM interactions sites are colored according to the interaction affected: priming = green, anchoring = orange, stabilizing = pink, scaffolding = white, regulating = yellow.

HCM variants therefore overlie distinct interaction sites when found in the context of the free or blocked head. As the same residue may be involved in more than one interaction, HCM variants have the potential to disrupt multiple IHM properties. Yet while MD and IHM residues overlap, particularly in the converter region, that 73 of 135 *MYH7* PVs and LPVs (54%) alter IHM interaction residues, as do all HCM variants in *MYL2* and *MYL3*, challenges the paradigm that these act primarily through the impairment of myosin motor functions (**Rayment et al., 1995**). As the IHM structure allows myosin to adopt the SRX state, we suggest that HCM PVs and LPVs that perturb dynamic IHM interactions would likely alter physiologic relaxation by the heart, and account for diastolic dysfunction, which is the major component of HCM pathophysiology.

## Distribution of DCM variants on IHM interactions

Parallel analyses of all DCM PVs and LPVs (n = 27) in *MYH7* (**Tables 3** and **5**, **Figure 3B** and **Figure 3—figure supplement 1B**) showed that these were over-represented at MD residues, but were not enriched in residues across all IHM interactions (p=0.65) nor among residues involved in interactions by the blocked head, free head, or tail (p≥0.25). Within specific components of IHM interactions (**Supplementary file 4a, b**), five DCM variants altered IHM priming (**Figure 3B**, green and **Video 6**), albeit with only nominal significance (p=0.019). Additional DCM variants altered blocked head residues involved in IHM stabilizing interaction (pink) and ELC scaffolding of blocked and free head residues (white), but these were not enriched over expectation. This data show that residues within IHM interaction sites are differentially impacted by variants that cause the distinct pathophysiologies of HCM and DCM.

Given the overall limited numbers of myosin DCM PV and LPVs, we also assessed two previously reported DCM *MYH7* PVs, S532P and F764L (**Schmitt et al., 2006**) that were not identified in the cardiomyopathy cohort studied here. S532P alters a blocked head residue (**Figure 3B**, side chain green), but is not involved in IHM interactions by the free head. F764L alters a blocked head residue that is involved in anchoring and a free head residue involved in IHM stabilization (**Figure 3B**, side chain orange and pink, respectively).

A far smaller proportion of myosin DCM PV and LPVs altered residues involved in IHM interactions (**Figure 1C**) than HCM variants. Together with the observed enrichment at MD functional domains shown above, this implies that DCM variants predominantly affect myosin MD properties,

**Table 2.** HCM likely pathogenic variants (LPVs) causing 95 MYH7 amino acid substitutions.

| Substitution | Location* | BH interaction* | Type* | FH interaction* | Type* | Δq‡ | Motor Domain Function§ |
|---|---|---|---|---|---|---|---|
| **MYH7** | **Motor Domain** | | | | | | |
| R17C | SH3 | h | Anchoring | | | -2 | |
| R143Q | | | | | | -2 | |
| K146N | | | | | | -1 | |
| R169G/K/S | Near P-loop | | | d.1 | Stabilizing | −2/−1/−2 | |
| Q193R | Near P-loop | | | | | +2 | |
| A199V | Near Loop 1 | | | | | 0 | |
| R204C | Loop 1 | | | | | -2 | Actin interface |
| G214D | Loop 1 | | | | | -1 | Actin interface |
| N232H | Near Switch 2 | | | | | +1 | |
| D239N | Switch 1 | | | | | +1 | ATP-Binding III |
| R243H† | Switch 1 | | | | | -1 | ATP-Binding III |
| F244C | Switch 1 | | | | | 0 | ATP-Binding III |
| K246I | Near Switch 1 | | | | | -1 | |
| F247L | Near Switch 1 | | | | | 0 | |
| I248T | Near Switch 1 | | | | | 0 | |
| G256E | Near Switch 1 | | | | | -1 | |
| I263M | Near ATP-Binding IV | | | | | 0 | |
| L267V | Near ATP-Binding IV | | | | | 0 | |
| Y283C | | | | | | 0 | |
| S291F | Near I-loop | | | | | 0 | |
| D309N | I-loop | | | | | +1 | MD-RLC interface |
| I323N | I-loop | | | | | 0 | MD-RLC interface |
| E328G | Near I-loop | | | | | +1 | |
| V338M | | | | | | 0 | |
| K351E | Near C-loop | | | | | -2 | |
| G354S | Near C-loop | | | | | 0 | |
| E374V | C-Loop | d.2 | Stabilizing | | | +1 | Actin interface |
| A381D | C-Loop | d.2 | Stabilizing | | | -1 | Actin interface |
| Y386C | Near C-loop | e | Stabilizing | | | 0 | |
| V406M | CM-loop | d.1 | Stabilizing | | | 0 | Actin Interface I |
| Y410D | CM-loop | d.1 | Stabilizing | | | -1 | Actin Interface I |
| V411I | CM-loop | d.1 | Stabilizing | | | 0 | Actin Interface I |
| L427M | | | | | | 0 | |
| T449S | Near Switch 2 | g | Priming | d.1 | Stabilizing | 0 | |
| R453S/H | Near Switch 2 | g | Priming | d.1 | Stabilizing | 0/−1 | |
| I457T | Near Switch 2 | | | | | 0 | |
| I478N | Near Switch 2 | | | | | 0 | |
| N479S | Near Switch 2 | | | | | 0 | |
| M493V/L/I | Relay | j | Anchoring | d.2 | Stabilizing | 0 | Relay |
| E497G/D | Relay | j | Anchoring | d.2 | Stabilizing | +1/0 | Relay |
| 1521T | Near Relay | | | | | 0 | |
| E536D | H-loop | f.2 | Priming | | | 0 | Actin Interface II |
| H576R | Loop 3 | | | | | +1 | Actin Interface III |

*Table 2 continued on next page*

*Table 2 continued*

| Substitution | Location* | BH interaction* | Type* | FH interaction* | Type* | Δq‡ | Motor Domain Function§ |
|---|---|---|---|---|---|---|---|
| G584R | Near Loop 3 | | | | | +2 | |
| V586A | Near Loop 3 | | | | | 0 | |
| R652G | Near Loop 2 | f.2 | Priming | | | -2 | |
| K657Q | Near Loop 2 | f.2 | Priming | | | -1 | |
| R663C | Near Loop 2 | f.2 | Priming | | | -2 | |
| R671C | Near SH2 helix | | | | | -2 | |
| R694C/H | SH2 helix | | | | | −2/–1 | |
| P710H/L | Converter | | | | | +1/0 | Converter |
| P731A | Converter | j | Anchoring | | | 0 | Converter |
| | | ELC-MHC | Scaffolding | | | | |
| G733E | Converter | j | Anchoring | d.2 | Stabilizing | -1 | Converter |
| | | ELC-MHC | Scaffolding | | | | |
| R739S | Converter | j | Anchoring | d.2 | Stabilizing | -2 | Converter |
| | | ELC-MHC | Scaffolding | | | | |
| K740N | Converter | j | Anchoring | d.2 | Stabilizing | -1 | Converter |
| | | ELC-MHC | Scaffolding | | | | |
| L749Q | Converter | j | Anchoring | | | 0 | Converter |
| | | ELC-MHC | Scaffolding | | | | |
| F758C | Converter | j | Anchoring | | | 0 | Converter |
| | | ELC-MHC | Scaffolding | | | | |
| V763M | Converter | | | d.2 | Stabilizing | 0 | Converter |
| G768R | Converter | | | | | +2 | Converter |
| L781P | Pliant | ELC-MHC | Scaffolding | ELC-MHC | Scaffolding | 0 | Pivot |
| R783H | Pliant | ELC-MHC | Scaffolding | ELC-MHC | Scaffolding | -1 | Pivot |
| | **Regulatory Domain** | | | | | | |
| R787C | Neck-ELC interface | ELC-MHC | Scaffolding | ELC-MHC | Scaffolding | -2 | |
| A797P | Neck-ELC interface | ELC-MHC | Scaffolding | ELC-MHC | Scaffolding | 0 | |
| L811P | Neck-ELC-RLC interface | ELC-MHC RLC-MHC | Scaffolding | RLC-MHC | Scaffolding | 0 | |
| | **S2** | | | | | | |
| K847E | Head-tail junction | | | | | -2 | |
| E848G | Head-tail junction | | | | | +1 | |
| M849T | Head-tail junction | | | | | | |
| A850D | Head-tail junction | | | | | -1 | |
| R858H | | g | Priming | g | Priming | -1 | |
| E894G | Ring 1 | f.1 | Priming | f.2 | Priming | +1 | |
| L898V | Ring 1 | f.1 | Priming | f.2 | Priming | 0 | |
| A901P | Ring 1 | f.1 | Priming | f.2 | Priming | 0 | |
| E903G | Ring 1 | f.1 | Priming | f.2 | Priming | +1 | |
| Q914H | | | | | | +1 | |
| D928N | Ring 2 | a | Stabilizing | a | Stabilizing | +1 | |
| E929K | Ring 2 | a | Stabilizing | a | Stabilizing | +2 | |
| E930Q | Ring 2 | a | Stabilizing | a | Stabilizing | +1 | |
| E949K | Ring 3 | | | | | +2 | |

*Table 2 continued on next page*

Alamo *et al.* eLife 2017;6:e24634. DOI: 10.7554/eLife.24634

*Table 2 continued*

| Substitution | Location* | BH interaction* | Type* | FH interaction* | Type* | Δq‡ | Motor Domain Function§ |
|---|---|---|---|---|---|---|---|
| E967K | | | | | | +2 | |
| R1053Q | | | | | | -2 | |
| | Light Meromyosin | | | | | | |
| E1356K | | | | | | +2 | |
| R1382Q | Near skip 2 residue | | | | | -2 | |
| L1428S | | | | | | 0 | |
| R1606C | Near skip 3 residue | | | | | -2 | |
| A1763T | | | | | | 0 | |
| R1781H | Near skip 4 residue | | | | | -1 | |
| M1782V | Near skip 4 residue | | | | | | |

All LPVs identified in 6112 HCM patients (**Walsh et al., 2017**) are shown. In MYH7 there 95 LPVs, leading to 95 distinct amino acid substitutions at 87 positions.

*__Supplementary file 1__ defines domain locations and **Supplementary file 2** defines IHM interactions, refined using the PISA analysis (**Krissinel and Henrick, 2007**) that are highlighted as in **Figure 3**, excluding the LMM variants. HCM LPVs located on the mesa are bolded.

†R243H was identified in both HCM and DCM cohorts. RLC–RLC denotes interface region between regulatory light chains.

‡Δq: variant induced electrical charge change. Δq defines whether the substituted amino acid changes the charge (bolded) or not (Δq = 0).

§Motor domain functions are ascribed only to variants within involved residues. Variants at IHM interactions sites are colored according to the interaction affected: priming = green, anchoring = orange, stabilizing = pink, scaffolding = white.

which is consistent with the predominant clinical manifestations of DCM *MYH7* variants - reduced contractile force but normal diastolic relaxation (*Lakdawala et al., 2012*).

## Charge change by HCM and DCM variants impacts the IHM

An electrostatic charge difference between the wild type and variant residue, $\Delta q$, is expected to further destabilize IHM inter- and intra-molecular interactions (*Tables 1–3*). Among HCM PVs that alter IHM interacting residues, 22/31 altered the charge (observed 71%, expected 49%, p=0.019): nine (29%) produced charge-gain ($+2 > \Delta q > 0$) and thirteen (42%) produced charge-loss ($-2 < \Delta q < 0$).

Charge changing HCM PVs are involved in multiple IHM interactions (*Table 1*). Four of five HCM PVs that alter IHM priming residues change the charge, including R870H, which is located on the S2 'kink' resulting in S2 bending which shapes the overall IHM (*Figure 3—figure supplement 2*, 'g' interaction). Nine of 12 PVs that alter IHM anchoring residues (*Figure 3—figure supplement 3*, 'i' and 'j') cause a charge change. Nineteen of 24 PVs (79%) change the charge of IHM stabilization residues, particularly those involved in 'd' interactions (*Figure 3—figure supplement 4*).

Both *MYL2* HCM PVs are charge changing and could destabilize MHC-RLC and RLC-RLC interactions (*Figure 3A*). *MYL2* R58Q resides in the loop between RLC helices B and C (*Figure 3—figure supplement 5*) that form the RLC-RLC interface. Spectroscopic studies of the RLC-RLC interface demonstrate that piperine destabilizes the SRX state, possibly in part by divalent cation binding (*Nogara et al., 2016b2016b*, *2016a*). As the E22K variant abolishes RLC phosphorylation and markedly reduces calcium-binding affinity (*Szczesna et al., 2001*), while the charge change associated with the R58Q variant would alter helical structures involved in the RLC-RLC interface, both variants should destabilize the SRX state. Consistent with this conclusion, ex vivo analyses of hearts from *MYL2* R58Q transgenic mice demonstrate higher ATPase rates and incomplete relaxation (*Greenberg et al., 2009*).

Five of seven (71%) DCM PVs and LPVs affecting IHM interacting residues also have electrostatic charge changes. Variant R904C should make the S2 Ring 1 more negative ($-2$) and strengthen IHM priming ('f.2') when the blocked head docks onto its S2, therein decreasing ATPase activity. By contrast, a nearby HCM variant (D906G *Figure 3—figure supplement 2*) makes the S2 Ring 1 more positive ($+1$) which weakens IHM priming ('f.2') with resultant increase in expected ATP turnover and numbers of released swaying free heads. Therefore the predicted reciprocal biophysical

**Table 3.** DCM pathogenic variants (PVs) and likely pathogenic variants (LPVs) causing 27 MYH7 amino acid substitutions.

| Substitution | Location[‡] | BH interaction[‡] | Type[‡] | FH interaction[‡] | Type[‡] | Δq[§] | Motor Domain Function[¶] |
|---|---|---|---|---|---|---|---|
| **MYH7** | **Motor Domain** | | | | | | |
| **G144V** | Near ATP Binding I | | | | | 0 | |
| G178R | ATP-binding II (P-loop) | | | | | +2 | ATP Binding II |
| G181R | ATP-binding II (P-loop) | | | | | +2 | ATP Binding II |
| **I201T** | Near 25/50 junction (Loop 1) | | | | | 0 | |
| R243H[†] | ATP-binding III (Switch 1) | | | | | -1 | ATP Binding III |
| G245E | ATP-binding III (Switch 1) | | | | | -1 | ATP Binding III |
| I248F | Near ATP-binding III | | | | | 0 | |
| R369Q | C-Loop (Loop 4) | d.2 | Stabilizing | | | -2 | Actin interface |
| D469Y | ATP-binding V (Switch 2) | | | | | +1 | ATP Binding V |
| I524V | Near H-Loop | f.2 | Priming | | | 0 | |
| **E525K** | Near H-Loop | f.2 | Priming | | | +2 | |
| I533V | H-Loop | f.2 | Priming | | | 0 | Actin interface II |
| R567H | Actin-interface III (Loop 3) | | | | | -1 | Actin interface III |
| N597K | Helix-loop-helix | | | | | +1 | Actin interface |
| C672F | Near SH2 Helix | | | | | 0 | |
| R783P | Pliant | ELC-MHC | Scaffolding | ELC-MHC | Scaffolding | -2 | Pivot |
| | **S2** | | | | | | |
| R904C*/H | Ring1 | f.2 | Priming | f.2 | Priming | −2/−1 | |
| E1152V | | | | | | +1 | |
| | **Light Meromyosin** | | | | | | |
| R1193H | Near skip 1 residue | | | | | -1 | |
| E1286K | | | | | | +2 | |
| R1434C | | | | | | -2 | |
| D1450N | | | | | | +1 | |
| R1574W | | | | | | -2 | |
| Q1794E | | | | | | -1 | |
| E1801K | Near skip 4 residue | | | | | +2 | |
| E1914K | | | | | | +2 | . |

All PVs and LPVs identified in 1315 DCM patients (**Walsh et al., 2017**) are shown. In *MYH7* there are 27 DCM pathogenic and likely pathogenic variants, leading to 27 distinct substitutions at 26 residues. DCM LPVs located on the mesa are bolded.

*R904C is the only DCM PV.

[†]R243H was identified in both HCM and DCM cohorts. In this cohort, no variants in *MYL2* or *MYL3* caused DCM.

[‡]**Supplementary file 1** defines domain locations and **Supplementary file 2** defines IHM interactions, refined using the PISA analysis (**Krissinel and Henrick, 2007**) that are highlighted as in **Figure 3**, excluding the LMM variants.

[§]Δq: variant induced electrical charge change. Δq defines whether the substituted amino acid changes the charge (bolded) or not (Δq = 0).

[¶]Motor domain functions are ascribed only to variants within involved residues. Variants at IHM interactions sites are colored according to the interaction affected: priming = green, stabilizing = pink, scaffolding = white.

consequences of adjacent *MYH7* variants on the IHM provide potential mechanisms to explain how nearby, indistinguishable groups variants with opposite charge changes cause the distinct pathophysiology of HCM and DCM (**Figure 1B** vs. C).

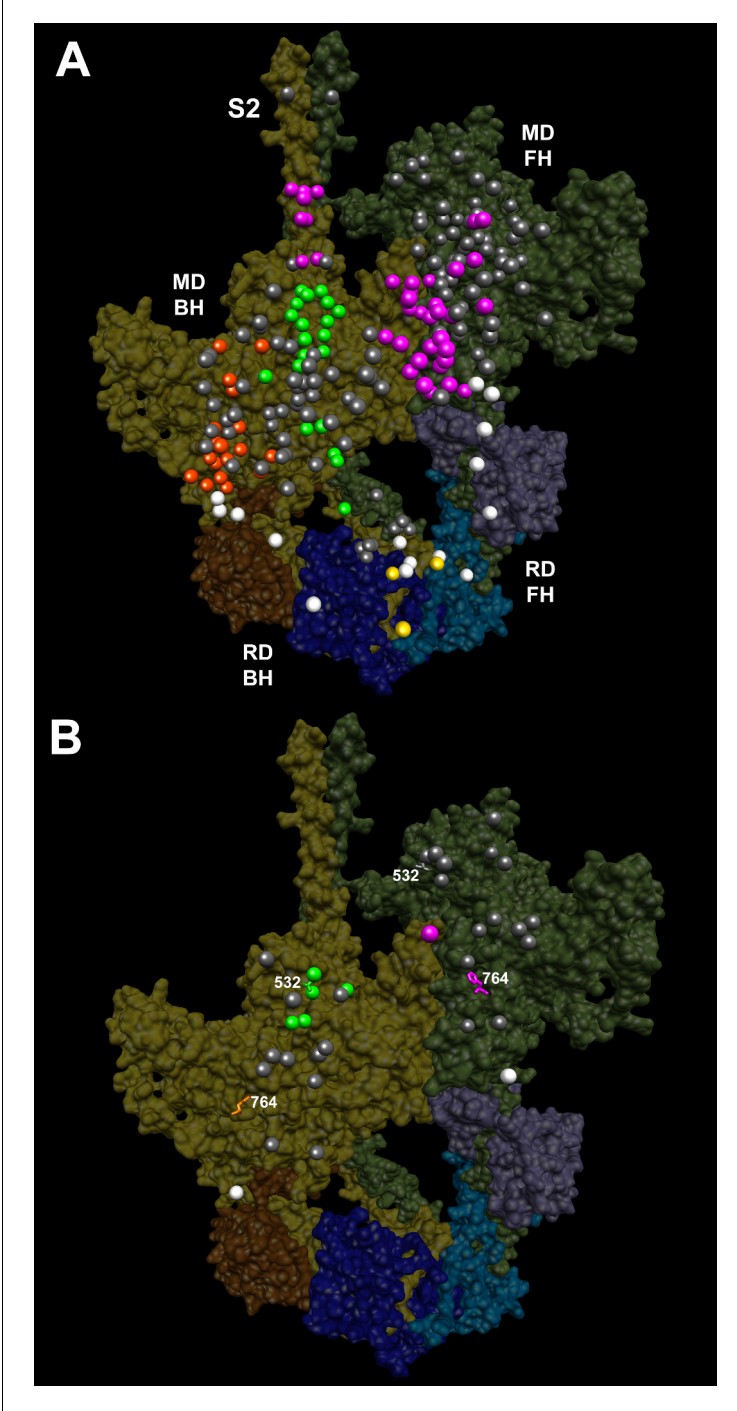

**Figure 3.** IHM PDB 5TBY models depicting pathogenic (PVs) and likely pathogenic variants (LPVs), in HCM and DCM (listed in *Tables 1–3*). Each variant appears as a pair, one located on or associated with the blocked head (BH, olive) and one on the free head (FH, green). Associated proteins are the essential light chain (ELC interacting with BH, brown; FH, purple) and regulatory light chain (RLC interacting with BH, dark blue; FH, light blue). (A) HCM PVs and LPVs that alter residues involved in IHM interactions (73/135 variants, 54%) are represented by colored balls: priming, green ('f' and 'g', *Figure 3—figure supplement 2*); anchoring, orange ('i' and 'j', *Figure 3—figure supplement 3*); stabilizing, pink ('a', 'd', and 'e', *Figure 3—figure supplement 4*); scaffolding, white (ELC-MHC and RLC-MHC); RLC-RLC interface, yellow (*Figure 3—figure supplement 5*). For variants with multiple IHM interactions, only one is depicted. PVs and LPVs that do not alter residues involved in IHM interactions are grey. (B) DCM PV and LPV (7/27; 26%) defined here are colored as described above, along with

*Figure 3 continued on next page*

*Figure 3 continued*

two prior PVs (S532P and F764L, denoted by side chains) that alter IHM interacting residues. Detailed IHM PDB 5TBY models of variants involved in specific interactions are provided in Supplemental Files and *Videos 5* and *6*).

The following figure supplements are available for figure 3:

**Figure supplement 1.** Wide-eye stereo pairs of the IHM PDB 5TBY model showing HCM pathogenic variants (**A**) and DCM pathogenic and likely pathogenic variants (**B**).

**Figure supplement 2.** Wide-eye stereo pair of the IHM PDB 5TBY model showing the five HCM variants that alter residues involved in IHM priming (intra-molecular interactions 'g' and 'f').

**Figure supplement 3.** Wide-eye stereo pair of the IHM PDB 5TBY model showing 13 HCM pathogenic variants involved in anchoring the IHM.

**Figure supplement 4.** Wide-eye stereo pair of the IHM PDB 5TBY model showing sites of 25 HCM variants that alter IHM stabilizing residues.

**Figure supplement 5.** Wide-eye stereo pair of the IHM PDB 5TBY model showing four variants located in the RLC-RLC interface region.

**Figure supplement 6.** Sequence alignment of human cardiac, chicken skeletal and tarantula striated MHC with the locations of HCM variants.

**Figure supplement 7.** Sequence alignment of human cardiac, chicken skeletal and tarantula striated MHC with the locations of DCM variants.

**Figure supplement 8.** Sequence alignment of human cardiac, chicken skeletal and tarantula striated ELC with the locations of HCM variants.

**Figure supplement 9.** Sequence alignment of human cardiac, chicken skeletal and tarantula striated RLC with the location of HCM variants.

## HCM variants on the mesa disrupt IHM interactions

The blocked myosin head structure in the IHM is similar to its pre-powerstroke structure (*Figure 2— figure supplement 2C* and *Video 4*) that contains a prominent flat myosin surface, denoted as the mesa (*Spudich, 2015*; *Spudich et al., 2016*; *Homburger et al., 2016*). The mesa is comprised of 277 aa, of which approximately half are IHM interacting residues. Based on the dynamic IHM

**Table 4.** The distribution of HCM variants across IHM and MD functional residues.

| Specified site | Variants within site | Rate | Amino acids within site | Expected rate | Rate ratio | P-value |
|---|---|---|---|---|---|---|
| IHM Interactions (All) | 73 | 0.541 | 447 | 0.2310 | 2.34 | 7.95e-15 |
| Priming | 17 | 0.126 | 113 | 0.0584 | 2.16 | 2.64e-03 |
| Anchoring | 24 | 0.178 | 156 | 0.0806 | 2.21 | 2.11e-04 |
| Stabilizing | 48 | 0.356 | 189 | 0.0977 | 3.64 | 5.37e-16 |
| Scaffolding | 24 | 0.178 | 120 | 0.0620 | 2.87 | 2.97e-06 |
| MD Functional | 39 | 0.289 | 194 | 0.1000 | 2.89 | 7.87e-10 |

The numbers of distinct pathogenic and likely pathogenic HCM variants (n = 135) affecting IHM interaction sites and motor domain (MD) functional residues are shown. Variant numbers are also tabulated separately for the four major IHM interactions: priming, anchoring, stabilizing and scaffolding. Note that a single variant may impact more than one interaction. The number of myosin amino acid residues (total protein length = 1935 residues) that compromise the IHM interaction sites or MD functions was used to determine the proportion of variants that would be expected to lie in the region of interest under the null (a uniform distribution), and the rates are compared with a binomial test. Full details of all variants are shown in *Tables 1* and *2*.

**Table 5.** The distribution of DCM variants across IHM and MD functional residues.

| Specified site | Variants within site | Rate | Amino acids within site | Expected rate | Rate ratio | P-value |
|---|---|---|---|---|---|---|
| IHM Interactions (All) | 7 | 0.259 | 447 | 0.2310 | 1.120 | 0.6550 |
| Priming | 5 | 0.185 | 113 | 0.0584 | 3.170 | 0.0186 |
| Anchoring | 0 | 0.000 | 156 | 0.0806 | 0.000 | 0.1650 |
| Stabilizing | 1 | 0.037 | 189 | 0.0977 | 0.379 | 0.5120 |
| Scaffolding | 1 | 0.037 | 120 | 0.0620 | 0.597 | 1.0000 |
| MD Functional | 7 | 0.259 | 210 | 0.1090 | 2.380 | 0.0222 |

The numbers of distinct pathogenic and likely pathogenic DCM variants (n = 27) affecting IHM interactions and motor domain (MD) functional residues are shown. Variant numbers are also tabulated separately for the four major IHM interactions: priming, anchoring, stabilizing and scaffolding. Note that a single variant may impact more than one interaction. The number of myosin amino acid residues (total protein length = 1935 residues) that compromise the IHM interaction sites or MD functions was used to determine the proportion of variants that would be expected to lie in the region of interest under the null (a uniform distribution), and the rates are compared with a binomial test. Full details of all variants are shown in *Table 3*.

structure, we expect that myosin can assume two mesa structures, corresponding to the blocked, docked head and the free head.

Fifty-two of 135 HCM variants (18 PVs and 34 LPVs; p=3.9e-12 versus expected), but only 3 DCM variants (p=0.79), reside on the mesa (*Tables 1–3*, *Supplementary file 5*, *Figure 4* and *Video 7*). Thirty-six HCM variants alter mesa residues that also participate in IHM interactions (4.1-fold above expectation, p=2.38e-13). When located on the blocked head, HCM mesa variants altered residues involved in priming (n = 9), anchoring and scaffolding (n = 15), and stabilizing (n = 6). When located on the free head HCM residues altered IHM stabilization (n = 21) and scaffolding (n = 1). By disrupting blocked S1 head interactions with S2 ('f.1' with S2 Ring 1 and 'g' with the S2 kink; *Figure 3—figure supplement 2*) that are crucial for establishing SRX, and stabilizing interactions between the blocked and free heads, HCM mesa variants could compromise relaxation and increase free heads available for force potentiation. The model that disruption of the IHM contributes to HCM pathophysiology is therefore entirely compatible with recent data suggesting an important role for the mesa. We expect that variants residing on the mesa (*Spudich, 2015*) and that are involved in IHM interactions (*Alamo et al., 2008*; *Jung et al., 2008*; *Luther et al., 2011*) may also impact MyBP-C interactions.

## Clinical interpretation of rare *MYH7* variants

Having identified a relationship between cardiomyopathy variants and residues identified as IHM interaction sites, we considered whether this information improved clinical variant interpretation. We compared the prevalence of all rare (ExAC global allele frequency <0.0001) missense variants in the 7427 cardiomyopathy

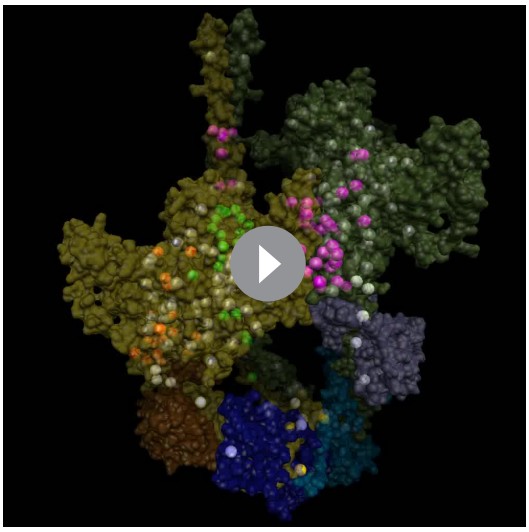

**Video 5.** Semi-transparent, space-filled PDB 5TBY IHM structure depicting the impact of HCM variants, when exposed to different IHM environments. HCM variants reside on the myosin blocked head (gold), myosin free head (olive), essential light chain (associated with blocked head (brown) or free head, (purple)), and regulatory light chain (associated with blocked head (dark blue) or free head (blue)). Variants are depicted as balls, colored according to the IHM interactions that are disrupted in each environment: priming, green; anchoring, orange; stabilizing, pink, scaffolding, white and regulatory, yellow. Grey variants alter residues that are not involved in IHM interactions. See legend of *Figure 3A*.

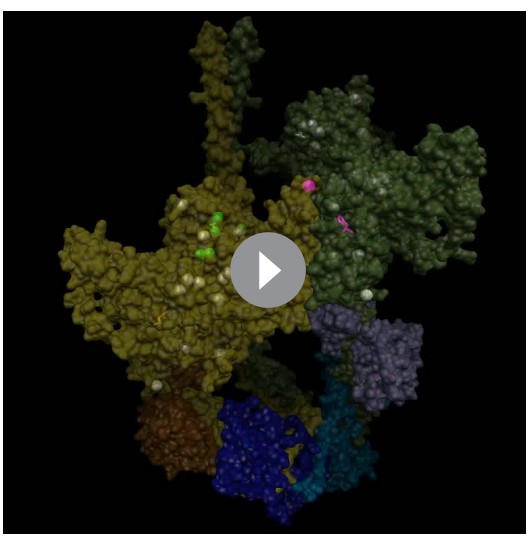

**Video 6.** Semi-transparent, space-filled PDB 5TBY IHM structure depicting the impact of DCM variants when residing on the myosin blocked head (gold) and myosin free head (olive). Variants are depicted as balls, colored according to the interactions that that disrupt: priming, green; anchoring, orange; stabilizing, pink; and scaffolding, white. Grey variants alter residues that are not involved in IHM interactions. The side chains of two previously reported DCM variants (*Schmitt et al., 2006*), S532P and F764L are also colored according to the IHM interaction that these disrupt. See legend of *Figure 3B*

subjects (*Walsh et al., 2017*) with 33,364 ancestry-matched ExAC subjects. We calculated disease odds ratios (under the assumption that each individual carries at most one rare variant, since individual-level genotypes are not available in ExAC, see Material and methods) and the etiologic fraction (EF, the proportion of variants in cases that are disease-causing), which represents the probability that a rare variant found on genetic testing in a cardiomyopathy subject is pathogenic (HCM, *Supplementary file 6*, DCM, *Supplementary file 7*). We calculated these measures for the whole protein, and for pre-specified sub-regions: the myosin head, MD functional sites, the mesa, a previously-described cluster of 29 amino acids in the converter region (*Homburger et al., 2016*), and the IHM interactions described here.

We identified 326 rare (global ExAC allele frequency <0.0001) variants in 854 of 6112 HCM patients (14%), resulting in 324 distinct aa substitutions at 300 residues. In the ExAC reference sample, 253 rare variants in 449 of 33,364 non-Finnish European subjects (1.4%), resulted in 252 distinct substitutions at 220 residues. Only 50 ExAC variants altered IHM residues (20%, expected 23%, p=0.23) and these were not enriched in specific interactions.

Regional analyses identified portions of the protein that are highly enriched in HCM, yielding a high probability that variants found in HCM patients are causative, with sufficient confidence for clinical application. As expected, these regions include the mesa (EF = 0.98) and MD functional sites (EF = 0.99), particularly the converter domain (EF = 0.99). Interpretive confidence is even higher for certain specific IHM interactions. Notably, rare variants involved in blocked head IHM priming ('g') and stabilizing ('d.1') residues were absent from ExAC but present in 62 HCM cases (p<e-200), while variants in five other IHM interactions were exceedingly rare in ExAC (EF ≥0.99). These regions accounted for ~44% of variants found in HCM cases (combined prevalence, 6%), but less than 4% of variants in ExAC (combined prevalence, 0.048%), indicating that variants within these sites have extraordinarily high prior probabilities of pathogenicity. Irrespective of our conclusion that these variants act by disrupting IHM interactions, these analyses robustly demonstrate that variants that alter a subset of *MYH7* residues are strongly predictive of HCM and clinically actionable.

Parallel comparisons of 58 rare variants in 68 of 1315 DCM patients (5.2%) indicated that a large proportion of those within the myosin head are pathogenic (EF = 0.75), but that this is insufficient to report novel variants as LPV without additional evidence of pathogenicity. Variants at functional MD residues have a high probability of pathogenicity (EF = 0.96). Among DCM rare variants that altered residues involved in IHM interactions (EF = 0.84), only those that altered IHM priming residues (EF = 0.96) achieved high probabilities of pathogenicity.

## Energetic consequences of SRX and DRX states

Experimental preparations of relaxed cardiac muscles (*Hooijman et al., 2011*) display two rates of ATP hydrolysis (slow turnover time, 138–144s; fast turnover time 20 s) that are respectively attributed to docked blocked heads and transiently docked free heads (the SRX state) and swaying free heads in the DRX state (*Alamo et al., 2016*). As such, the proportion of myosin heads in the SRX or DRX state impacts cardiac muscle expenditure of ATP and energy efficiency, parameters that are critical for continuous pumping.

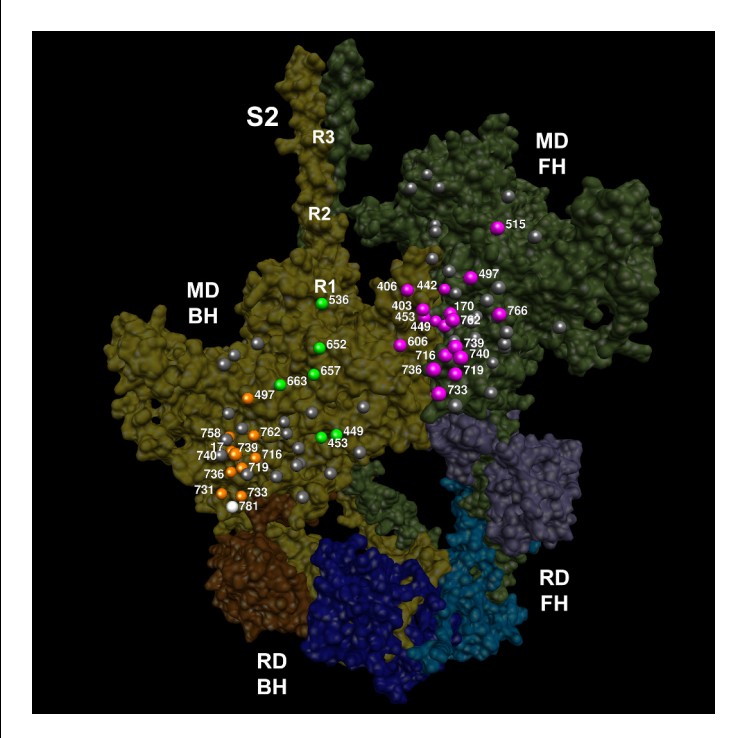

**Figure 4.** The location of 14 variants located on the mesa (*Spudich, 2015*; *Homburger et al., 2016*) of the blocked head (BH, olive) and free head (FH, green). The three negatively-charged rings in the blocked head S2 are labeled R1, R2 and R3. The associated sarcomere proteins are depicted as in *Figure 3*. The myosin mesas are roughly orthogonal (blocked head mesa, parallel to the page; free head mesa, perpendicular to the page; see *Video 7*). HCM PVs are enriched on the mesa and in IHM interactions when located either on the blocked or free head, suggesting that these disrupt crucial determinants of cardiac relaxation, accounting for diastolic dysfunction in HCM.

We calculated a theoretical estimate of the populations of relaxed cardiac myosin heads in DRX or SRX (Material and methods) based on the assumptions that (a) the IHM structure accounts for the SRX and that (b) activation of cardiac muscle only impacts free heads in the DRX state while docked blocked heads remain parallel to the thick filament. The latter assumption is supported by fluorescence analyses of RLC in relaxed rat heart fibers, that show both parallel and perpendicular alignment to the thick filament (*Kampourakis and Irving, 2015*) and by evidence that cardiac myosin heads remain in the SRX state with tetanic fiber activation (*Hooijman et al., 2011*). Based on these assumptions, the asymmetric configuration of myosin heads in the cardiac IHM could yield savings of one ATP molecule when the free head is docked to blocked head (SRX) that would be lost when swaying free heads increase ATP turnover (DRX, *Figure 1A*). As such HCM variants in *MYH7*, *MYL2*, and *MYL3* that impair IHM interactions or that destabilize the RLC-RLC interface (*Figure 3—figure supplement 5*) such as occurs with RLC phosphorylation, would reduce SRX:DRX proportions, consume more ATP, and promote free head swaying with potential myosin-actin interactions and force production.

## Discussion

Biophysical and biochemical analyses of the sarcomere's chemomechanical cycle have defined the molecular basis of force production by striated muscles and illuminated how this is perturbed by cardiomyopathy mutations. Identification of dynamic IHM intra- and inter-molecular interactions that produce populations of relaxed myosins in DRX and SRX states provides additional fundamental insights into sarcomere biology that informs the molecular pathogenesis of HCM and DCM.

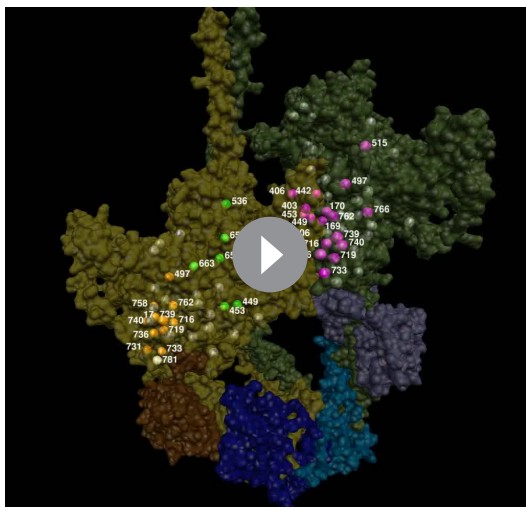

**Video 7.** Semi-transparent, space-filled 5TBY IHM structure depicting only HCM variants that reside on the blocked head and free head mesas. The myosin blocked head (gold), myosin free head (olive), essential light chain (associated with blocked head (brown) or free head (purple)), and regulatory light chain (associated with blocked head (dark blue), or free head (blue)) are shown. Mesa variants (numbered according to the amino acid that is altered) are depicted as balls, colored according to the IHM interactions that is disrupted in each environment: priming, green; anchoring, orange; stabilizing, pink, and scaffolding, white. Grey variants alter residues that are not involved in IHM interactions. See legend of *Figure 4*.

We demonstrate that similar proportions of HCM and DCM variants (26–29%) impact myosin residues with MD functions, albeit with distinctly different distributions. DCM variants were enriched only in the nucleotide-binding site (p=0.00019) but were notably absent from the converter domain, a region where residues also function in IHM interactions. By contrast, HCM variants are highly enriched in the converter domain (p=1.25e-8) and on the mesa (p=3.89e-12), domains that both participate in MD functions (HCM variant enrichment, p=7.9e-9) and IHM interactions (HCM variant enrichment, p=7.95e-15). Indeed, HCM variants were 1.19-fold enriched in residues involved only in MD functions, 1.74-fold enriched in residues involved only in IHM interactions, and 4.17-fold enriched in residues that participate in both, providing an account for why HCM PVs and LPVs, unlike DCM variants, have broad effects throughout the contraction and relaxation phases of the cardiac cycle (*Figure 1B*).

While the location of HCM variants within functional MD residues of myosin, in vitro motility assays, and single molecule biophysical studies (*Tyska et al., 2000*; *Spudich, 2014*; *Spudich et al., 2016*) provide mechanisms for abnormal contractility, these do not account for the prevalent and profound relaxation deficits that characterize HCM. By contrast, evidence that HCM variants in *MYH7, MYL2,* and *MYL3* alter residues that support the dynamic properties of IHM provide mechanisms by which these disrupt cardiac relaxation and energetics (*Figure 1A*, *Figure 3—figure supplement 5*; see Material and methods). HCM variants that impair IHM interactions would reduce SRX:DRX proportion, promote additional myosin-actin interactions, and consume more ATP, resulting in altered relaxation, energetics and force production.

Previous studies of selected cardiomyopathy variants, that employed early thick filament and IHM structures, suggested that these might impact IHM formation (*Blankenfeldt et al., 2006*; *Alamo et al., 2008*; *Moore et al., 2012*). Our study advances these conclusions in several ways. Capitalizing on a novel human β-cardiac myosin IHM PDB 5TBY structure we considered if cardiomyopathy residues altered specific IHM interactions. Second, we studied a comprehensive and unselected set of PVs and LPVs that were independently identified in a large cardiomyopathy cohort (7427 HCM and DCM patients) and compared these to variants in 33,370 ExAC controls. We found that only HCM variants were significantly enriched in many IHM interacting residues, most notably in those that stabilize the IHM (p=5.37e-16 vs. expectation). As most HCM variants resulted in a charge change (*Tables 1* and *2*), these are expected to adversely affect IHM interactions. Taken together, these data provide strong support that HCM variants perturb cardiac relaxation by disrupting IHM interactions. A corollary to this conclusion is that the IHM interacting residues identified by the PDB 5TBY structure are physiologically valid.

We recognize several limitations in our analyses. First, there are neither atomic nor near-atomic structures of the human cardiac IHM. Although a near-atomic (0.6 nm) structure of the *Lethocerus* thick filament backbone has been reported (*Hu et al., 2016*), the resolution for the IHM is comparable (2.0 nm) to the tarantula structure employed here, perhaps because the intrinsic flexibility of swaying free heads limits higher resolution (*Brito et al., 2011*; *Sulbarán et al., 2013*). Second, using homologous *MYH7* sequences, we converted the tarantula (PDB 3JBH) into a human (PDB 5TBY)

IHM model, which fits well into the human beta-cardiac 3D-reconstruction (*Figure 2a*). While capitalizing on the high evolutionary conservation between human MYH7 and tarantula striated myosin (60% amino acid identity), especially residues involved in IHM interactions (*Alamo et al., 2016*), we expect that future models will refine the resolution of interactions identified in the PDB 5TBY model. Third, the tarantula striated muscle lacks MyBP-C and titin, molecules that may interact with IHMs to form helices that protrude from vertebrate thick filaments and limit interactions between myosin heads and actin (*Alamo et al., 2008*; *Jung et al., 2008*; *Luther et al., 2011*). The locations of several HCM variants hint that these may have additional functional consequences that impair interactions between consecutive IHMs. Fourth, while our data support the conclusion that the structural states defined by the IHM are responsible for the DRX and SRX functional states, these cannot be proven to be synonymous.

We also expect that high-resolution human cardiac thick filament structures will extend these insights to other HCM genes and variants. MyBP-C harbors many HCM variants that reduce protein levels (*Harris et al., 2011*) and evoke functional consequences similar to MyBP-C phosphorylation: increase calcium sensitivity of active force and acceleration of rigor force and rigor stiffness development (*Kunst et al., 2000*). The C1C2 fragment of MyBP-C (*Gruen and Gautel, 1999*) is predicted to interact with the second of three negatively charged rings (*Blankenfeldt et al., 2006*) on the myosin S2 domain (*Figure 2A* and *Figure 4*, labeled as R1-3), and Ring 2 residues participate in stabilizing the IHM. Charge inverting (+2) variants on Ring 2 (E924K and E930K in *Table 1* and *Figure 3—figure supplement 4* 'a'; E927K and E935K, described in (*Blankenfeldt et al., 2006*) could disrupt MyBP-C binding (*Lee et al., 2015*), and destabilize the IHM, as would HCM variants that deplete MyBP-C protein levels. Moreover, destabilization of the IHM could account for the significant reduction of SRX that occurs in mice lacking MyBP-C (*McNamara et al., 2016*). With the enrichment of *MYH7* variants in IHM interacting residues reported here, we suggest that altered physiologic ratios of SRX and DRX populations is the mechanism by which HCM disrupts cardiac relaxation.

## Integrated consequences of cardiomyopathy variants on myocardial physiology

HCM and DCM are dominant disorders and sarcomeres contain an admixture of normal and mutated myosin proteins. As *MYH7* DCM variants primarily impact functional MD residues (e.g. nucleotide binding and hydrolysis), these will depress contractility in proportion to the summed force produced by myosins with and without variants (*Schmitt et al., 2006*; *Spudich et al., 2016*), with concurrent reduction in ATP consumption. The resultant under-performance of sarcomeres containing DCM variants promotes compensatory ventricular dilation (Frank-Starling mechanism) to maintain circulatory demands (*Figure 1C*).

By contrast, (*Schmitt et al., 2006*; *Spudich et al., 2016*) HCM variants impact both MD functions and IHM properties. Any pair of myosins involved in asymmetric IHM interactions may contain zero, one, or two mutated residues with singular or dual consequences when located on the blocked or free head (*Tables 1* and *2*, *Figure 3*). The 135 HCM PVs and LPVs described here will alter 39 functional MD residues on each head and 73 IHM interacting residues, located on the blocked or free heads or S2 domain (*Supplementary file 3f*). Notably, 41 of these 135 variants altered only one IHM interaction, while 32 altered residues involved in at least two interactions simultaneously, typically a different one for each head (*Tables 1* and *2*). We suggest that the effects of HCM variants on contractility, their consequences on MD functions, and the multiple effects on IHM interactions, confers a complexity that can explain the dissimilar results obtained when myosin mutations are studied in the context of single myosin molecules that only assess MD functions and myofibril experiments that also interrogate the IHM (*Moore et al., 2012*). For example, R453C can simultaneously affect IHM priming ('g' interaction) when located on the blocked head (*Figure 3—figure supplement 2*) and stabilization ('d.1') when on the free head (*Figure 3—figure supplement 4*). Biomechanical assays of myosin R453C shows decreased actin-sliding velocity and ATPase, but increased force production (*Sommese et al., 2013*). These dichotomous results may be explained by the IHM paradigm (*Figures 1B* and *3A*). Located near the switch 2 with the nucleotide-binding pocket, R453C would impair S1 functions that are assayed by single molecule in vitro analyses. When located on the blocked head ('g', *Figure 3—figure supplement 2*) the charge change of R453C ($\Delta q = -2$) would destabilize blocked myosin heads and increase the number of heads (the released blocked head and the associated partner free heads) that are available for force potentiation. As such, the increase in

sarcomere force observed with many HCM variants likely reflects the combined effect on MD functions and IHM interactions that increase myosin populations in DRX and reduce those in SRX states.

HCM variants will further impact cardiac physiology because of the relationship between the proportion of myosins in the DRX and SRX states and energy consumption (*Figure 1B*). Swaying free myosin heads in the DRX state consume 5-fold greater ATP than in the SRX state (*Hooijman et al., 2011*), and physiologic SRX:DRX levels during relaxation may reduce energy consumption and support life-long continuous activity. By disrupting IHM interactions, HCM variants would reciprocally increase the proportion of myosins in the DRX state and decreased the proportion in the SRX state – resulting in increased ATP consumption and metabolic demands, yielding energy deficiencies that characterize human HCM hearts (*Ashrafian et al., 2003*). These observations also provide a plausible mechanistic explanation for why MYK-461, an ATPase inhibitor, improved the pathophysiology in HCM mouse models (*Green et al., 2016*).

In summary, the integrated interpretation of HCM and DCM variants on myosin MD functions and dynamic IHM interactions provides a fuller understanding of sarcomere biology in health and disease. While DCM variants evoke primary deficits in MD functions (*Figure 1C*), HCM variants perturb both MD functions and IHM interactions (*Figure 1B*). By disrupting IHM priming, anchoring, stability, and scaffolding interactions, HCM variants decrease populations of docked myosin heads (SRX) and increase swaying heads (DRX) that are available for force production, thereby promoting hypercontractility. The reduction of myosin populations in SRX would perturb diastolic relaxation and increase ATP consumption, causing energy deficiencies in HCM hearts (*Ashrafian et al., 2003*). Notably, these are primary biophysical deficits produced directly by HCM variants that are unrelated to secondary compensatory changes observed in other heart diseases. Clinical observations support this conclusion: hyperdynamic contraction, diastolic dysfunction (*Ho et al., 2002*), and energy deficiency (*Jung et al., 1998*; *Crilley et al., 2003*) precedes hypertrophic remodeling in young HCM variant mutation carriers. We conclude that the identification of sarcomere protein residues involved in relaxation provide new opportunities to target the IHM to improve HCM pathophysiology and other heart diseases with diastolic dysfunction. By extension of these analyses to skeletal myopathies we anticipate further insights into critical roles of the highly conserved IHM in the relaxation of striated muscles.

## Material and methods

### Overview
We first generated a new homologous human β-cardiac myosin IHM quasi-atomic model, and used this to identify residues at sites of IHM interactions. These interaction sites were predefined from structural models, without prior knowledge of the location or distributions of HCM or DCM variants. We then examined the locations of a large series of clinically defined cardiomyopathy variants (*Walsh et al., 2017*). All these independently classified HCM and DCM PVs or LPVs were included in our statistical analyses. Finally, for predefined *MYH7* functional regions we compared the total prevalence rare variants (not stratified by clinical consequence) in cardiomyopathy cases and a reference population to derive unbiased and quantitative estimates of clinical interpretability. Code to reproduce statistical analyses (in R) and figures (in Chimera) is available on GitHub (*Ware, 2017*).

### Myosin II MHC, ELC and RLC sequences
Sequences include human β-cardiac MHC (P12883), ELC (P08590) and RLC (P10916); chicken skeletal muscle MHC (P13538) and tarantula (*Aphonopelma sp.*) skeletal muscle ELC (GenBank KT390185), RLC (PDB KT390186) (*Zhu et al., 2009*) and MHC (GenBank KT619079) (*Alamo et al., 2016*). Alignments of the human β-cardiac, chicken skeletal and tarantula striated MHC, ELC and RLC that are shown in *Figure 3—figure supplements 6–9* were generated with Clustal (*Larkin et al., 2007*) over the JalView platform (*Waterhouse et al., 2009*).

### Homology modeling and rigid docking
Human β-cardiac myosin ELC, RLC and MHC (MD and S2) were modeled as described (*Alamo et al., 2016*). The human β-cardiac myosin interacting-heads motif (*Figure 2A* and *Video 1*) has been deposited into the Research Collaboratory for Structural Bioinformatics Protein Data Bank

(PDB 5TBY). A rigid fitting of this model was done against the human cardiac negatively stained thick filament 2.8 nm resolution 3D-reconstruction (EMD-2240) (*Al-Khayat et al., 2013*) using the Chimera 'Fit in Map' option (*Figure 2A* and *Video 1*) (*Pettersen et al., 2004*).

## SAXS analysis

Sample preparation and data collection details for squid HMM are previously reported (*Gillilan et al., 2013*). Scattering intensity, I(q), is in arbitrary units, with q = 4π sin(θ)/λ in units of Å$^{-1}$. The wavelength λ = 1.2563 Å and the scattering angle is 2θ as measured from the direct beam. Scattering profiles for PDB 5TBY and tarantula PDB 3JBH were computed using both FoXS (*Schneidman-Duhovny et al., 2013*) and CRYSOL (*Svergun et al., 1995*) to compare agreement, especially at scattering angles q > 0.3 Å$^{-1}$. As described previously, CRYSOL required more than the default number of harmonics (*Svergun et al., 1995*) for agreement with FoXS in the WAXS regime and the algorithms agreed on the location and magnitude of the difference between the PDB models despite a minor baseline shift (not shown). Currently neither of these methods model flexibility, an effect that could also introduce baseline shifts at wider angles. The FoXS calculations reported here fit both the computed profiles to the squid HMM experimental data using the default parameters of hydration layer, excluded volume, and background adjustment (*Gillilan et al., 2013*). In the case of *Figure 2B,C* the 'profile offset' optimization was performed to obtain best fit with the data at widest angles (*Schneidman-Duhovny et al., 2013*). The goodness of fit parameter χ, reported by FoXS, is defined as

$$\chi = \sqrt{\frac{1}{M}\sum_{i=1}^{M}\left(\frac{I_{exp}(q_i) - I(q_i)}{\sigma(q_i)}\right)^2},$$

where $I_{exp}(q_i)$ is the experimental scattering profile measured at M points $(q_i)$ with error given by $\sigma(q_i)$ Model profiles, $I(q_i)$, that agree with the data to within expected noise levels give χ values close to unity. (*Gillilan et al., 2013*; *Schneidman-Duhovny et al., 2013*; *Svergun et al., 1995*)

## Identification of IHM interaction sites

The analysis of the interactions was done as described (*Alamo et al., 2016*), using the 'Protein interfaces, surfaces and assemblies' (PISA) service at the European Bioinformatics Institute (http://www.ebi.ac.uk/pdbe/prot_int/pistart.html) (*Krissinel and Henrick, 2007*). Interactions were described as intra- or inter-molecular according to the customary convention used in IHM studies (*Woodhead et al., 2005*; *Alamo et al., 2016*).

## Accession numbers

The atomic coordinates of the human β-cardiac myosin IHM quasi-atomic model have been deposited into the Research Collaborators for Structural Bioinformatics as PDB entry 5TBY.

## Curation of HCM and DCM variants

Variants found in samples from 7427 individuals referred for genetic testing with a referral diagnosis of HCM or DCM, sequenced in accredited diagnostic laboratories with clinical-grade variant interpretation (3826 individuals sequenced at Oxford Medical Genetics Laboratories (*Walsh et al., 2017*), and 3668 individuals from the Partners HealthCare Laboratory of Molecular Medicine (*Pugh et al., 2014*; *Alfares et al., 2015*) were used in our analysis. In total, 6112 HCM patients had *MYH7* sequencing, including 4185 also with *MYL2* and *MYL3* sequencing, while 1315 DCM cases had *MYH7* sequencing and 543 DCM also with *MYL2* and *MYL3* sequencing. First, all curated variants reported as pathogenic by either laboratory (HCM PV, *Table 1*) were assessed in the mapping analyses (*Figure 3A*, *Supplementary file 3a*, *Figure 3—figure supplements 1–5*). Variants reported as likely pathogenic (HCM LPV, *Table 2*, *Supplementary file 3b*, *Figure 3A*) were then analyzed for replication. Subsequent analyses of specific interactions include HCM PVs and LPVs combined (*Table 4*, *Supplementary file 3a-d*). Given the smaller number of variants in DCM, and that PV and LPV behaved similarly in HCM, all DCM PV and LPV were considered together (*Tables 3* and *5*, *Supplementary file 4*, *Figure 3B*).

## Assessing the regional distribution of variants

To determine whether variants were over-represented in a region of interest (*Supplementary files 1*, *2*), the proportion of variants (*Supplementary files 3–5*) that fell within the region was compared with the proportion that would be expected under a uniform distribution (given by the length of the region / total protein length). Proportions were compared using a binomial test, implemented in R (*Ware, 2017*).

## *MYH7* variants in the general population

In order to confirm the importance of the pre-specified IHM interaction sites (*Supplementary file 2*), and to determine the strength of the disease association for variants at different sites, we compared the total prevalence of rare variants in the case series described above (irrespective of clinical laboratory classification) with the prevalence amongst reference samples from the Exome Aggregation Consortium (ExAC; *Lek et al., 2016*); *Supplementary files 6*, *7*). Rare (ExAC global AF <0.0001) missense variants were extracted from the ExAC vcf (version 0.3.1). Analysis was restricted to the ExAC sub-population of European ancestry, as the case series is drawn from a predominantly Caucasian background (*Walsh et al., 2017*). The median number of ExAC samples genotyped at each variant site was used as the ExAC population size (33,364), and prevalence was compared using the binomial test (*Ware, 2017*). A disease odds ratio (OR) was estimated (under the assumption that each rare variant was found in a distinct individual), and the etiological fraction (EF) was calculated as (OR – 1)/OR (*Ware, 2017*). The EF, a commonly used method in epidemiology, estimates the proportion of affected variant carriers in which the variant caused the disease, which can also be interpreted as the probability that an individual rare variant, found in a proband, is responsible for the disease (*Greenland and Robins, 1988*; *Robins and Greenland, 1989*; *Cole and MacMahon, 1971*; *Walsh et al., 2017*).

## Mapping of the MHC, ELC and RLC HCM and DCM variants to the human β-cardiac myosin IHM structure PDB 5TBY

The sites of 43 curated HCM aa substitutions (*Table 1*), 95 LPV substitutions (*Table 2*), and 27 DCM aa substitutions (PVs plus LPVs, *Table 3*) are shown in *Figure 3* and *Figure 3—figure supplements 1–5*. UCSF Chimera software version 10.2 (*Pettersen et al., 2004*) was used to build *Figures 2A*, *3* and *4* and *Figure 3—figure supplements 1–5*. The Chimera sessions (.py) used for making *Figure 2A* and *Figure 2—figure supplement 2*, *Figure 3* and *Figure 3—figure supplements 1–5* and *Figure 4* are available in GitHub (*Ware, 2017*). A copy is archived at https://github.com/elifesciences-publications/eLife_Alamo2017.

## Assessment of charge changes

In order to determine what proportion of missense changes might be expected to be charge-changing, we determined all possible single nucleotide variant events for MYH7 (1935 codons x nine possible SNV codon changes = 17,415 events). For 12,850 possible missense substitution events, we then determined the charge change: 2681 would lead to a negative charge change, 3604 to a positive charge change, and 6565 would be charge neutral, yielding an expectation that 49% of missense substitutions might be charge changing.

## Estimates of SRX and DRX populations in relaxed skeletal and cardiac muscle

### Relaxed skeletal muscle

The proportion of skeletal muscle myosins in the SRX or DRX states has been deduced from the population of heads parallel or perpendicular to the filament axis, using fluorescence polarization from bifunctional rhodamine probes bound to the RLC. In relaxed rabbit psoas muscle, approximately 70% of all myosin heads are parallel to the fiber axis, while ~30% of all myosin heads are perpendicular (*Fusi et al., 2015*), implying that the head population comprises 50% blocked heads that are all parallel, 20% docked free heads that are parallel and 30% swaying free heads that are perpendicular. According to the relaxed IHM state model for skeletal muscle (*Alamo et al., 2016*), the blocked head remains docked while the free head sways away for a time $t_s$ and is docked only for a time $t_d$. Therefore for a total time $t = t_s + t_d$, the swaying duty cycle $D_s$ is defined as the

ratio $t_s/t$. $D_s$ can be estimated in the relaxed skeletal muscles from measurements of the RLC orientation using fluorescence polarization from bifunctional rhodamine probes bound to the skeletal RLC and cardiac RLC (Fusi et al., 2015) (*Kampourakis and Irving, 2015*). In relaxed demembranated rabbit psoas muscle ~70% of the heads have their RLC orientation roughly parallel to the fiber axis, similar to those in the IHM structure, while the RLC orientation of the remainder ~30% of heads is more perpendicular, which is inconsistent with the IHM structure (Fusi et al., 2015). For the estimation of $D_s$ and the SRX/DRX ratio we assume a population of N myosin molecules, so as each myosin molecule has one free (FH) and one blocked (BH) head, the sub-populations of free ($N_{FH}$) and blocked ($N_{BH}$) heads sum $N_{BH} + N_{FH} = 2N$. The populations of heads measured in the parallel conformation is $N_{par} = N_{BH\text{-}par} + N_{FH\text{-}par}$, while the populations of heads measured in the perpendicular conformation is $N_{per} = N_{BH\text{-}per} + N_{FH\text{-}per}$. In the relaxed state, blocked heads are docked so $N_{BH\text{-}per} = 0$, therefore $N_{per} = N_{FH\text{-}per}$. As a result the number of heads in the perpendicular conformation $N_{per} = N_{FH\text{-}per} = (t_s/t)N_{FH} = D_s N_{FH} = D_s N$ while the number of free heads in the parallel conformation is $N_{FH\text{-}par} = N_{FH} - N_{FH\text{-}per} = N_{FH} - (t_s/t)N_{FH} = N_{FH} - D_s N_{FH} = (1-D_s)N$. Therefore the number of heads in the parallel conformation is $N_{par} = N_{BH\text{-}par} + N_{FH\text{-}par} = N_{BH\text{-}par} + (1 - D_s)N$, and as $N_{BH\text{-}par} = N$ because blocked heads are docked in the relaxed state, then $N_{par} = (2 - D_s)N$; in contrast the number of heads in the perpendicular conformation is $N_{per} = D_s N$. Therefore the relative number of heads in the parallel and perpendicular conformations are respectively $N_{par}/2N = (2 - D_s)N/2N = (2 - D_s)/2$ and $N_{per}/2N = D_s N/2N = D_s/2$. Finally the ratio R of heads in the parallel and perpendicular conformation is $R = N_{par}/N_{per} = (2 - D_s)/D_s$ so we conclude that $D_s = 2/(1+R)$. Therefore in skeletal muscle R = 0.70/0.3 = 2.3 so $D_s$ = 0.60. Assuming that FHs spend 60% of the cycle released and swaying, and 40% docked on the blocked head the estimated ratio of skeletal muscle heads in the SRX and DRX states is SRX/DRX = $(2 – D_s)/D_s$ = 2.3, predicting an estimated 2.3-fold reduction of ATP hydrolysis. As skeletal muscle myosin heads rapidly transition out of SRX into DRX, such as occurs with tetanic activation (*Hooijman et al., 2011*) (*Brito et al., 2011*), the changing population of myosin SRX and DRX heads in skeletal muscle achieves energy-saving imposed by the IHM structure but retains the ability to rapidly shift myosin into DRX for force potentiation.

## Relaxed cardiac muscle

Assessment of the population of cardiac myosin heads based on parallel or perpendicular alignment to the filament axis is more complicated because fluorescence analyses of RLC in relaxed ventricular fibers from rat hearts show both orientations (*Kampourakis and Irving, 2015*). However, because previous studies show that cardiac myosin heads remain in the SRX state with tetanic fiber activation (*Hooijman et al., 2011*), we deduced that activation of cardiac muscle only impacts free heads in the DRX state; docked blocked heads in the SRX state are not recruited. Supporting this idea, there is rapid re-ordering and presumably re-forming of the IHM following cardiac contraction, a finding that suggests that the IHM must be regenerated during the relaxation phase that follows every heartbeat (*Huxley et al., 1982*) (*Yuan et al., 2015*). Based on the assumption that IHM and SRX states are one and the same in cardiac muscle, we calculated the corresponding $D_s$ and SRX/DRX ratios from the fluorescent analysis results of (*Kampourakis and Irving, 2015*) that showed both parallel and perpendicular orientations. By further assuming that the myosin parallel and perpendicular populations are roughly similar, a conservative estimate of cardiac $D_s$ is 80% and the SRX/DRX ratio is 1.5. This conservative estimate of Ds = 0.8 was considerable higher than skeletal muscle, that is, most free heads are not interacting with the blocked head, with a smaller SRX/DRX of 1.5. Based on these observations, we suggest that cardiac IHM swaying duty cycle $D_s$ should be considerable higher than skeletal muscle, that is, most free heads are not interacting with the blocked head. With these assumptions, the asymmetric configuration of myosin heads in the cardiac IHM would yield savings of one ATP molecule when the free head is docked to blocked head (SRX) that would be lost when swaying free heads increase ATP turnover (DRX, *Figure 1A*). Moreover, HCM variants in *MYH7*, *MYL2*, and *MYL3* that impair IHM regulating interactions, like phosphorylation-dependent destabilization of the RLC-RLC interface (*Figure 3—figure supplement 7*) would reduce SRX:DRX proportions and consume more ATP and promote additional myosin-actin interactions and force.

## Acknowledgements

We thank Dr. Neal Epstein for initial suggestions regarding HCM variants on interaction 'a', Dr. Tom Kirchhausen for early help in this project, Dr. Hind AL-Khayat for permission to use the human cardiac thick filament 3D-map (EMD- 2240), and Dr. Gustavo Márquez for his manuscript assistance. This work was supported by the Leducq Foundation (11-CVD-01; CES and JGS), the Wellcome Trust (107469/Z/15/Z; JSW), the MRC (JSW), NHLBI (HL084553; JGS) and the Howard Hughes Medical Institute (CES and RP). Centro de Biología Estructural del Mercosur (www.cebem-lat.org) (to RP) Cornell High Energy Synchrotron Source (CHESS) is supported by the NSF and NIH/NIGMS via NSF award DMR-1332208, and the MacCHESS resources are supported by NIGMS award GM-103485 (to REG). We dedicate this paper to the memory of Dr. Guillermo Whittembury.

## Additional information

### Competing interests

JGS, CES: is a founder and owns shares in Myokardia Inc., a startup company that is developing therapeutics that targets the sarcomere. The other authors declare that no competing interests exist.

### Funding

| Funder | Grant reference number | Author |
| --- | --- | --- |
| National Institutes of Health | NHLBI-HL084553 | Jonathan G Seidman |
| Howard Hughes Medical Institute | | Christine E Seidman |
| Howard Hughes Medical Institute | | Raúl Padrón |
| Wellcome Trust | 107469/Z/15/Z | James S Ware |
| Medical Research Council | | James S Ware |
| National Institutes of Health | GM-103485 | Richard E Gillilan |
| Foundation Leducq | 11CVD-01 | Jonathan G Seidman<br>Christine E Seidman |

The funders had no role in study design, data collection and interpretation, or the decision to submit the work for publication.

### Author contributions

LA, Performed research, Analyzed data, Wrote the paper with critical input from all authors; JSW, CES, Designed research, Performed research, Analyzed data, Wrote the paper with critical input from all authors; AP, Performed research, Analyzed data ; REG, Performed research, Analyzed data; JGS, Designed research, Performed research, Analyzed data ; RP, Designed research, Performed research, Analyzed data, Wrote the paper with critical input from all authors

### Author ORCIDs

Lorenzo Alamo, http://orcid.org/0000-0003-3893-2631
James S Ware, http://orcid.org/0000-0002-6110-5880
Antonio Pinto, http://orcid.org/0000-0002-0665-9872
Richard E Gillilan, http://orcid.org/0000-0002-7636-3188
Jonathan G Seidman, http://orcid.org/0000-0002-9082-3566
Christine E Seidman, http://orcid.org/0000-0001-6380-1209
Raúl Padrón, http://orcid.org/0000-0002-1412-2450

## Additional files

### Supplementary files

• Supplementary file 1. Domains of human $\beta$-cardiac myosin.

• Supplementary file 2. Intra- and inter-molecular interactions sequences involved in human β-cardiac myosin interacting-heads motif (IHM) PDB 5TBY.

• Supplementary file 3. HCM variants cluster on residues involved in IHM-related inter- and intra-molecular interactions.

• Supplementary file 4. DCM-causing variants cluster in distinct regions of *MYH7* from HCM-causing variants.

• Supplementary file 5. Variants Clustered on the Myosin Mesa.

• Supplementary file 6. Comparison of prevalence of rare (ExAC global AF $<1\times10^{-4}$) missense variants in *MYH7* in 6112 HCM cases and ExAC controls.

• Supplementary file 7. Leveraging regional distribution for the clinical interpretation of DCM-causing variants.

### Major datasets

The following dataset was generated:

| Author(s) | Year | Dataset title | Dataset URL | Database, license, and accessibility information |
|---|---|---|---|---|
| Lorenzo Alamo, James S Ware, Antonio Pinto, Richard E Gillilan, Jonathan G Seidman, Christine E Seidman, Raúl Padrón | 2017 | Human Beta Cardiac Heavy Meromyosin Interacting-Heads Motif Obtained by Homology Modeling (Using Swiss-Model) of Human Sequence from Aphonopelma Homology Model (PDB-3JBH), Rigidly fitted to Human Beta-Cardiac Negatively Stained Thick Filament 3D-Reconstruction (EMD-2240) | http://www.rcsb.org/pdb/explore/explore.do?structureId=5TBY | Publicly available at the RCSB Protein Data Bank (accession no. 5TBY) |

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
