## [Decision Letter]

Thank you for submitting your article "Effect of Myosin Variants on Interacting-Heads Motif Explain Distinct Hypertrophic and Dilated Cardiomyopathy Phenotypes" for consideration by *eLife*. Your article has been reviewed by three peer reviewers, and the evaluation has been overseen by a Reviewing Editor and Harry Dietz as the Senior Editor. The following individuals involved in review of your submission have agreed to reveal their identity: Sharlene Day (Reviewer #1); Anne Houdusse (Reviewer #3).

The reviewers have discussed the reviews with one another and the Reviewing Editor has drafted this decision to help you prepare a revised submission. In general, all reviewers and the Reviewing Editor found the study very interesting but some concerns were raised that need to be resolved. With important modifications, in particular by recognizing the limits of the current knowledge of the interactions in the IHM and the difficulties and inherent limitations of evaluating how a single mutation can impact interactions and function, the article will be of interest for people in the field since there has not been a global description of how the current model of the IHM interactions can result in HCM or DCM disease.

Summary:

In an attempt to understand the functional consequences of different hypertrophic and dilated cardiomyopathy mutations in human cardiac β myosin heavy chain, the manuscript integrates data from protein modeling using a quasi-atomic model, mutations found in the patient population and protein-protein interactions of the head region in an attempt to establish mechanistic bases for structure-function. Major conclusions are reached concerning the structural pathogenesis of the hypertrophic and dilated cardiomyopathy mutations utilizing a large cardiomyopathy database comprised of 7,427 individuals. The authors use a low resolution model of the interacting heads motif that was previously published to globally identify which residues mutated in these diseases are involved in stabilization interactions. The data show that the hypertrophic and dilated cardiomyopathic variants differentially affect the motor domain and interacting heads motif of the protein, potentially informing the mechanistic interpretations of how the different mutations influence sarcomeric function and power development at the molecular level.

Essential revisions:

1) A major concern on the part of multiple reviewers was that the relatively low resolution of the published data, and the paucity of rigorously defined interactive residues are not commensurate with the "definite" tone taken by the authors at multiple places in the manuscript when stating their conclusions. That is, there is too much emphasis throughout the manuscript on trying to establish dogma and generalize conclusions. For example, in paragraph three of the Discussion section, the authors speculate that mutations in titin and MYBPC3 could also impact IHM properties and alter relaxation. The majority of mutations in these 2 genes are truncating mutations that are hypothesized to act primarily by loss of function. And since the residues in MYBPC3 and titin that interact with S2 are not known, it's difficult to predict which missense variants would affect these interactions. Proposing that disruption of IHM formation or stability is a common mechanism for mutations in all thick filament genes for HCM seems to be a bit of a leap. It is not clear from the current paragraph, subsection “HCM variants implicated in MyBP-C binding disrupt IHM interactions”, whether the authors here mostly report on what has been previously proposed or whether they have themselves performed a study that agrees with previous reports. In particular, they should report whether the EM model can provide a proposition on the residues likely involved in interaction with MyBP-C and if it is restricted to the second charged ring on the S2. In fact, the impact of MyBP-C on the IHM should also be considered. While cMyBP-C may stabilize the IHM (Zhogbi, Woodhead, Moss and Craig, 2008) and SRX state (McNamara, et al., 2016) the sites of interaction are still highly debated. The structure proposed by Alamo et al. is based on a small subset of contacts in the literature and highly speculative at best. The authors might consider removing this section to avoid further confusion.

The reviewers raise the very legitimate question as to why, in previous manuscripts, the authors state that their previous models didn't have the resolution needed to draw such conclusions. What has led to their changing their minds? In the absence of a convincing argument that addresses this concern, they have simply used a large database to formulate hypotheses that now can be rigorously tested. The limitations of the work presented in this manuscript need to be clearly and explicitly stated in the Discussion.

2) Alamo et al. state "Although there is limited resolution of the cardiac interconnecting inter-molecular interactions between sequential IHMs along the helices of myosin heads protruding from the thick filament backbone, we note that 12 HCM PVs and LPVs are also positioned to alter inter-IHM interactions data not shown) and limit swaying free heads from actin interactions." Can the authors clarify what they believe the resolution to be for these inter-IHM and intra-IHM interactions? Would the inter- and intra-IHM contacts be affected by the fact that the tarantula striated muscle IHM quasi-atomic cryo-EM structure comes from a thick filament with a 4 start helix while the human cardiac thick filament is a 3 start helix?

3) The manuscript makes the assumption that IHM first observed in smooth muscle and then in tarantula muscle is the structural state responsible for the SRX kinetic state first described by Cooke, 2011, in skeletal and then cardiac muscles. While I realize this assumption may be correct, can the authors point to clear experimental evidence showing that IHM and SRX states are one and the same in cardiac muscle as it pertains to the literature?

4) SAXS and model (Figure 2): The authors compare theoretical scattering curves from their homology model and experimental scattering curve from squid HMM (subsection “Comparisons of the human β-cardiac myosin IHM quasi-atomic PDB 5TBY with existing models”). In order to better compare the two curves, the authors should provide a χ2 and residual curves (resulting from the comparison of the two scattering curves). The authors do not demonstrate with this comparison whether different possible models that could be built within the EM map would be distinguished. The limitation of the model for defining precisely the contact regions within the IHM should be discussed further. The sentence “Although resolution of the IHM structure as derived by cryo-EM does not define specific atomic contacts or individual side chain densities, this model nonetheless identifies molecular interactions.”, is too ambiguous to indicate how confident one can be on the position of various regions that the authors describe as being involved in IHM contacts. For example, what is really known about what forms the RLC/RLC interface (subsection “Charge Change by HCM and DCM variants Impacts the IHM” third paragraph, on MYL2 R58Q)? Is there sufficient resolution to indicate which residues form the interface between these two side chains?

5) The mutations studied here are not clearly identified from the figures and tables. The statistics mentioned in the text (for example subsection “HCM variants altering IHM interactions”) cannot always allow the identification of the specific mutations they correspond to – as an example, in paragraph three: which are the five of eight variants involved in'i' interactions that affect residues that also participate in another IHM interaction? The authors should better introduce how the mutations were classified, which mutations belong with which category – rather than providing% on whether or not they belong to a particular subclass. This could be improved possibly by the figure legend of the table and a clearer reference to the table in which the residues beyond these numbers are presented. It seems important also to describe how many mutations do not fit in these conclusions to evaluate the limits of the prediction this study can provide.

6) Energetic consequences of SRX and DRX states: This part of the result doesn't really exploit much of the authors' model or mutations. It is not clear what the authors are trying to add here that has not been discussed in previous publications or would be relevant with what is presented above. In any case, for the demonstration/calculations/ definition of Ds, the authors should explicitly say that all the parameters and calculations are based on an assumption: the blocked head remains in a parallel configuration and that only SRX or DRX configurations of the myosin is possible (one head blocked / the other one freely able to dock or stay free).

The authors should provide substantial support for this important assumption that the blocked head always stays in the parallel configuration. In any case, this model doesn't seem appropriate here when mutations that would destabilize the blocked head are considered.

All the parameters introduced here do not really make a point relevant about DCM or HCM mutations that justify their Introduction in the paper, as currently written.

---

## [Author Response]

*Essential revisions:*

*1) A major concern on the part of multiple reviewers was that the relatively low resolution of the published data, and the paucity of rigorously defined interactive residues are not commensurate with the "definite" tone taken by the authors at multiple places in the manuscript when stating their conclusions.*

The reviewer comments bring out an important potential source of confusion that we have attempted to clarify in the revised manuscript. While the quasi-atomic model presented here, PDB 5TBY, is consistent with existing low-resolution structural data, its strength lies in the multiple lines of additional evidence that justify the appropriateness of the physiologically important IHM interactions studied here:

1) There is substantial evolutionary conservation between human β cardiac myosin and striated myosin from the tarantula (60% amino acid identity), especially among residues involved in IHM interactions (Alamo et al., 2016).

2) The PDB 5TBY model fits well to the 28Å resolution 3D-reconstruction for the negatively stained human cardiac thick filament (EMD-2240) (Al-Khayat et al., 2013) and closely matches crystal structures of fragments for human β-cardiac S1 motor domain (PDB 4DB1) and S2 fragments (PDB 2FXM) reported by Blankenfeldt and colleagues (Blankenfeldt et al., 2006).

3) Most importantly, we remind the reviewers that the IHM interacting residues reported in Alamo et al. (Alamo et al., 2016) were independently predefined without knowledge of the human genetic variants reported by Walsh et al., 2016, that were classified by clinical laboratories. We also considered all reported variants (both pathogenic and likely pathogenic) identified in HCM, DCM, and those within the general population in this study. To clarify this we have added an overview to the Materials and methods that states:

“These interaction sites were predefined from structural models (see Supplemental Information), without prior knowledge of the location or distributions of HCM or DCM variants. We then examined the locations of a large series of clinically defined cardiomyopathy variants (Walsh et al., 2016). All these independently classified HCM and DCM PVs or LPVs were included in our statistical analyses.”

Predicated on these elements we provide a comprehensive approach that demonstrates:

a) IHM interacting residues are strongly and specifically enriched for HCM-associated variants. We show that this enrichment is independently significant for pathogenic and likely pathogenic HCM variants, providing both initial and replicated observations. As the prototypic clinical manifestations of HCM are closely aligned with the predicted functional consequences of altering IHM structures, we are confident that this association reflects cause and effect.

b) The location of interacting residues improves the discrimination of pathogenic HCM variants from benign variants.

c) That IHM interacting residues are NOT enriched for DCM variants nor variants found in the general population. If the replicated HCM finding were spurious, we would have seen a signal in these other variant datasets.

Even with our confidence in these results, in this revision we provide readers with statements to alert them that the model is not perfect, and we specify limitations in the Discussion. In the Results, we introduce the model with a disclaimer that states:

“The IHM structure as derived by cryo-EM does not define specific atomic contacts or individual side chain densities”

In the Discussion we have added a paragraph that specifies limitations of our analyses.

“We recognize several limitations in our analyses …..”

Including that “….we expect that future models will refine the resolution of interactions identified

*That is, there is too much emphasis throughout the manuscript on trying to establish dogma and generalize conclusions. For example, in paragraph three of the Discussion section, the authors speculate that mutations in titin and MYBPC3 could also impact IHM properties and alter relaxation. The majority of mutations in these 2 genes are truncating mutations that are hypothesized to act primarily by loss of function. And since the residues in MYBPC3 and titin that interact with S2 are not known, it's difficult to predict which missense variants would affect these interactions. Proposing that disruption of IHM formation or stability is a common mechanism for mutations in all thick filament genes for HCM seems to be a bit of a leap. It is not clear from the current paragraph, subsection “HCM variants implicated in MyBP-C binding disrupt IHM interactions”, whether the authors here mostly report on what has been previously proposed or whether they have themselves performed a study that agrees with previous reports. In particular, they should report whether the EM model can provide a proposition on the residues likely involved in interaction with MyBP-C and if it is restricted to the second charged ring on the S2. In fact, the impact of MyBP-C on the IHM should also be considered. While cMyBP-C may stabilize the IHM (Zhogbi, Woodhead, Moss and Craig, 2008) and SRX state (McNamara, 2016) the sites of interaction are still highly debated. The structure proposed by Alamo et al. is based on a small subset of contacts in the literature and highly speculative at best. The authors might consider removing this section to avoid further confusion.*

We have revised the entire text to provide specific rather than general conclusions. We agree that our data does not define specifically identify interactions involved in MYPBC3 interactions, and as recommended we have removed this section from the Results. However, because prior publications implicate MyBPC interactions with three negative rings on the myosin S2, the discussion includes a speculation that MyBPC3 variants could abrogate these interactions and destabilize the IHM, with adverse effects on SRX (subsection “Energetic Consequences of SRX and DRX States”). This speculation is supported by published studies that demonstrate biochemical evidence for reduced SRX in *MYBPC3*- null mice (McNamara et al., 2016).

*The reviewers raise the very legitimate question as to why, in previous manuscripts, the authors state that their previous models didn't have the resolution needed to draw such conclusions. What has led to their changing their minds? In the absence of a convincing argument that addresses this concern, they have simply used a large database to formulate hypotheses that now can be rigorously tested. The limitations of the work presented in this manuscript need to be clearly and explicitly stated in the Discussion.*

Our comments on the “resolution of the model” are addressed above. The most compelling reason for our “change of mind” is that human genetics now provides an independent strategy that validates these interactions. That only HCM variants are selectively enriched in IHM interactions (unlike variants in DCM patients or in the general population) and that most HCM variants alter the charge of the encoded residue provides complementary supportive information. We explicitly state this in the Discussion section:

“Taken together, these data provide strong support that HCM variants perturb cardiac relaxation by disrupting IHM interactions. A corollary to this conclusion is that the IHM interacting residues identified by the PDB 5TBY structure are physiologically valid.”

As requested, we have added a detailed limitation section to the Discussion (paragraph five).

*2) Alamo et al. state "Although there is limited resolution of the cardiac interconnecting inter-molecular interactions between sequential IHMs along the helices of myosin heads protruding from the thick filament backbone, we note that 12 HCM PVs and LPVs are also positioned to alter inter-IHM interactions data not shown) and limit swaying free heads from actin interactions." Can the authors clarify what they believe the resolution to be for these inter-IHM and intra-IHM interactions? Would the inter- and intra-IHM contacts be affected by the fact that the tarantula striated muscle IHM quasi-atomic cryo-EM structure comes from a thick filament with a 4 start helix while the human cardiac thick filament is a 3 start helix?*

We elected to delete this paragraph, which was not central to our analyses or conclusions. However, for the interest of the reviewers we provide the follow explanation. The tarantula thick filament is perfectly helical and has 4 helices of IHMs on the backbone surface. On each helix, there are adjacent IHMs (with intermolecular interactions "b" and "c") that are spaced every 14.5 nm producing a "crown", of IHMs that are rotated by 30 degrees, from the prior crown, so that three crowns occur every 43.5 nm. In tarantula thick filaments, the IHMs on each of these three crowns are equivalent (making a 43.5 nm repeat) that interact through intermolecular interactions "b" and "c" (described in Alamo, et al., 2016). The initial text stated “we note that 12 HCM PVs and LPVs are also positioned to alter inter-IHM interactions” because these variants impact "b" and "c" interactions.

*3) The manuscript makes the assumption that IHM first observed in smooth muscle and then in tarantula muscle is the structural state responsible for the SRX kinetic state first described by Cooke, 2011in skeletal and then cardiac muscles. While I realize this assumption may be correct, can the authors point to clear experimental evidence showing that IHM and SRX states are one and the same in cardiac muscle as it pertains to the literature?*

We are not aware of any experimental evidence that definitely shows that IHM and SRX states are one. Indeed, we expect that new technologies would need to be developed to simultaneously assess structure (e.g. IHM interactions) and function (e.g., ATP cycling). While cryo-EM may be able to provide multiple structural states that differ in conformation within a single specimen this only provides the structural correlate of kinetic states – and cannot directly prove that a structure equals a functional state.

To remind the reader of these issues, we have modified the text to indicate we assume the IHM structure accounts for the SRX as follows:

“We calculated a theoretical estimate of the populations of relaxed cardiac myosin heads in DRX or SRX (Supplemental Information) based on the assumptions that a) the IHM structure accounts for the SRX and that…..”

“Fourth, while our data support the conclusion that the structural states defined by the IHM are responsible for the DRX and SRX functional states, these cannot be proven to be synonymous. “

*4) SAXS and model (Figure 2): The authors compare theoretical scattering curves from their homology model and experimental scattering curve from squid HMM (subsection “Comparisons of the human β-cardiac myosin IHM quasi-atomic PDB 5TBY with existing models”). In order to better compare the two curves, the authors should provide a χ2 and residual curves (resulting from the comparison of the two scattering curves).*

As requested by the reviewer, the SAXS curve in Figure 2 has been revised to include an additional panel (2C) showing the relative deviation of the predicted scatter of 5TBY from the experimental squid HMM data. Additionally, we have provided a dashed line showing the relative deviation between the two models, 5TBY and 3JBH on the same scale for comparison. The legend of Figure 2 has been updated, adding the new part, 2C. A corresponding comment in the Results now states: “The “χ” goodness of fit parameters reported by the FoXS program are 1.5 for 3JBH and 1.85 for 5TBY respectively (Schneidman-Duhovny et al., 2013). Relative deviation between the 5TBY and 3JBH calculated scattering profiles (dashed line) falls below the noise level of the experimental data (red line), and consequently the two models cannot be distinguished. “

Quality of fit parameters “χ” as reported by the FoXS software have been included in the Results. It should be noted that it has become customary in the SAXS literature to report χ rather than χ2. The two are related in the obvious way, and we have added the formula and a reference in the Material and Methods section.

*The authors do not demonstrate with this comparison whether different possible models that could be built within the EM map would be distinguished.*

The aim of this comparison was only to show that the human cardiac IHM PDB 5TBY quasi-atomic model fit the only available IHM experimental SAXS profile (for squid) not comparing the predicted SAXS profiles of different human cardiac IHM models that could be built as we only have this one.

*The limitation of the model for defining precisely the contact regions within the IHM should be discussed further. The sentence “Although resolution of the IHM structure as derived by cryo-EM does not define specific atomic contacts or individual side chain densities, this model nonetheless identifies molecular interactions.”, is too ambiguous to indicate how confident one can be on the position of various regions that the authors describe as being involved in IHM contacts. For example, what is really known about what forms the RLC/RLC interface (subsection “Charge Change by HCM and DCM variants Impacts the IHM” third paragraph, on MYL2 R58Q)? Is there sufficient resolution to indicate which residues form the interface between these two side chains?*

Please see responses to comment #1. As indicated above, we now provide limitations in the Discussion.

5) The mutations studied here are not clearly identified from the figures and tables.

We apologize for this confusion. To address this we have done the following:

1) Edited the text to indicate that all variants reported by Walsh and colleagues were studied.

“All sarcomere PVs and LPVs previously identified in 6112 HCM and 1315 DCM cases (Walsh et al., 2016) were analyzed.”

2) Added details to the Materials and methods:

“We then examined the locations of a large series of clinically defined cardiomyopathy variants (Walsh et al., 2016). All these independently classified HCM and DCM PVs or LPVs were included in our statistical analyses.”

3) Provided Table 1–Table 3 as part into the main text, as these tables report the specific variants and charge change, variant locations in motor domains and IHM interactions when the variant is located on the blocked or free head.

Walsh and colleagues (Walsh et al., 2016) used standard clinical criteria to classify pathogenic and likely pathogenic variants in HCM and DCM patients. We did not reiterate the details in this published manuscript, but provide an overview in Materials and methods (subsection “Curation of HCM and DCM variants”).

*The statistics mentioned in the text (for example subsection “HCM variants altering IHM interactions”) cannot always allow the identification of the specific mutations they correspond to – as an example, in paragraph three: which are the five of eight variants involved in'i' interactions that affect residues that also participate in another IHM interaction? The authors should better introduce how the mutations were classified, which mutations belong with which category – rather than providing% on whether or not they belong to a particular subclass.*

We apologize for the confusion. Throughout the manuscript, statistical analyses are provided for all IHM interactions, and when specifically noted, for specified IHM interactions, to highlight interactions that are particularly enriched for HCM variants. In this revision we provide a summary table for HCM (Table 4) and DCM (Table 5) that describe statistical enrichment for all variants across the IHM and for major specific interactions, so that readers can readily access these essential data. In addition to presenting statistical summaries, we have also added to the legend a reference to the source table, where the full listing and annotations for the variants included in that particular analysis.

We apologize for our intended meaning was unclear. The revised text has clarified this sentence.

“Twenty-four HCM variants (Figure 3, orange and Figure 3—figure supplement 3) altered blocked head residues involved in IHM anchoring interactions (p=2.1e-4). Five of eight anchoring variants (involved in “i” interactions; Figure 3—figure supplement 3) affect residues that also participate in another IHM interaction (Table 1, Table 2) we cannot discern whether one or both interactions are more relevant to HCM pathogenesis. “

*This could be improved possibly by the figure legend of the table and a clearer reference to the table in which the residues beyond these numbers are presented. It seems important also to describe how many mutations do not fit in these conclusions to evaluate the limits of the prediction this study can provide.*

As indicated above, our conclusions are based analyses of all HCM and DCM PVs and LPVs that were identified in 6112 HCM and 1315 DCM cases (Walsh, 2016 #48). No variants were excluded from any analyses reported here. We have modified the legends for Table 1–Table 3 to indicate this. Furthermore, for each table presenting statistical summaries we have added to the legend a reference to the source table listing the variants included in that particular analysis.

*6) Energetic consequences of SRX and DRX states: This part of the result doesn't really exploit much of the authors' model or mutations. It is not clear what the authors are trying to add here that has not been discussed in previous publications or would be relevant with what is presented above. In any case, for the demonstration/calculations/ definition of Ds, the authors should explicitly say that all the parameters and calculations are based on an assumption: the blocked head remains in a parallel configuration and that only SRX or DRX configurations of the myosin is possible (one head blocked / the other one freely able to dock or stay free).*

*All the parameters introduced here do not really make a point relevant about DCM or HCM mutations that justify their Introduction in the paper, as currently written.*

We agree that the original section of energetic consequences is predicated on multiple assumptions and for cardiac muscle represents a theoretic calculation. However we respectfully disagree that these data are irrelevant to HCM and DCM variants. As indicated in the text HCM variants are known to cause increased energy requirements (Ashrafian et al., 2011). This is one of three central disease components (along with hypercontractility and impaired relaxation) that has been heretofore unexplained.

To balance these issues, we have substantially modified the text as follows. The manuscript now contains a brief overview of energetic consequences that occur by altering IHM interactions (starting in paragraph two of subsection “HCM variants on the mesa disrupt IHM interactions“) and populations of myosins in SRX or DRX. The Supplemental Information provides details on the assumptions, specific parameters, and all calculations used to estimate these populations, and how varying population sizes relates to energy consumption.

*The authors should provide substantial support for this important assumption that the blocked head always stays in the parallel configuration. In any case, this model doesn't seem appropriate here when mutations that would destabilize the blocked head are considered.*

The energetics section in the Supplemental Information specifically address this:

“However, because previous studies show that cardiac myosin heads remain in the SRX state with tetanic fiber activation (Hooijman et al., 2011), we deduced that activation of cardiac muscle only impacts free heads in the DRX state; docked blocked heads in the SRX state are not recruited. Supporting this idea, there is rapid re-ordering and presumably re-forming of the IHM following cardiac contraction, a finding that suggests that the IHM must be regenerated during the relaxation phase that follows every heartbeat (Huxley et al., 1982) (Yuan et al., 2015) […] By further assuming that the myosin parallel and perpendicular populations are roughly similar […]”